# Attention Projection Mixing with Exogenous Anchors

**Jonathan Su** [1]

## Abstract

Cross-layer reuse of early attention projections can improve optimization and data efficiency, but it creates a structural conflict: the first layer must simultaneously act as a stable, reusable anchor for all deeper layers and as an effective computational block. We demonstrate that this tension constrains the performance of internal-anchor designs. We propose ExoFormer, which resolves the conflict by learning *exogenous anchor projections* outside the sequential layer stack. We introduce a unified normalized mixing framework that mixes queries, keys, values, and gate logits using learnable coefficients (exploring coefficient granularities: elementwise, headwise, and scalar), and we show that normalizing anchor sources is key to stable reuse. ExoFormer variants consistently outperform their internal-anchor counterparts, and the dynamic variant yields $\sim 1.5$ downstream accuracy points while matching validation loss using $\sim 1.5\times$ fewer tokens than Gated Attention. We explain this efficacy via an *Offloading Hypothesis*: external anchors preserve essential token identity, allowing layers to specialize exclusively in feature transformation. We release code and models to facilitate future research.

## 1. Introduction

The Transformer architecture (Vaswani et al., 2017) has become the foundation for modern large language models (LLMs) and a wide array of sequential and contextual tasks. A core component of its success is the multi-head self-attention mechanism, which enables dynamic, context-dependent interactions across sequences. However, as models scale in depth to capture more complex abstractions, ensuring stable and efficient training alongside effective information propagation remains a challenge.

[1]Independent Researcher. Correspondence to: Jonathan Su <270985@learning.gsis.edu.hk>.

*Proceedings of the $43^{rd}$ International Conference on Machine Learning*, Seoul, South Korea. PMLR 306, 2026. Copyright 2026 by the author(s).

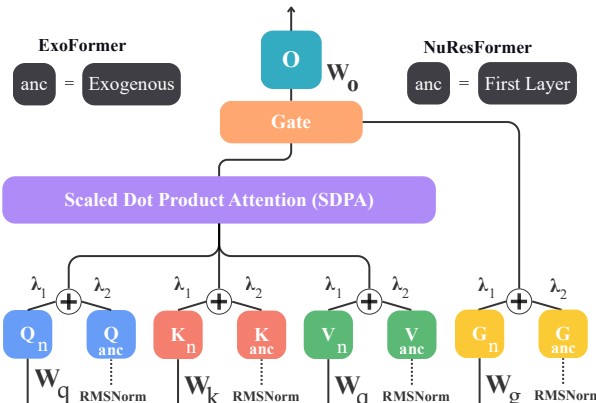

*Figure 1.* For **NuResFormer**, the anchor components are sourced from the first layer. For **ExoFormer**, they are provided by dedicated exogenous projections. All residualized sources are normalized before being mixed with the current layer's projections, and the scaled dot product attention (SDPA) output is gated.

Studies show that token-level information can become diluted in deeper layers, a phenomenon linked to over-smoothing (Shi et al., 2022; Zhou et al., 2021). This has spurred interest in more direct mechanisms for preserving early representations.

However, existing solutions largely operate in isolation and leave key questions unanswered. ResFormer (Zhou et al., 2025) focuses solely on residualizing *values*, leaving the potential reuse of other attention components (queries, keys, and gating logits) unexplored.

We introduce a unified framework for cross-layer mixing across all attention pathways through systematic evaluation of the contribution of mixing queries ($Q$), keys ($K$), and gating logits ($G$), in addition to the established value ($V$) residual using different coefficient granularities (elementwise, headwise, scalar). A key insight is that applying RMSNorm to the residual sources before mixing resolves distributional mismatch, enabling stable and beneficial reuse.

Analyzing architectures that reuse first-layer projections reveals a fundamental tension. The first layer is forced to serve two roles: (1) as a stable, reusable anchor for all deeper layers and (2) as an effective computational block for progressive feature transformation. This dual objective inherently limits effectiveness in both roles.

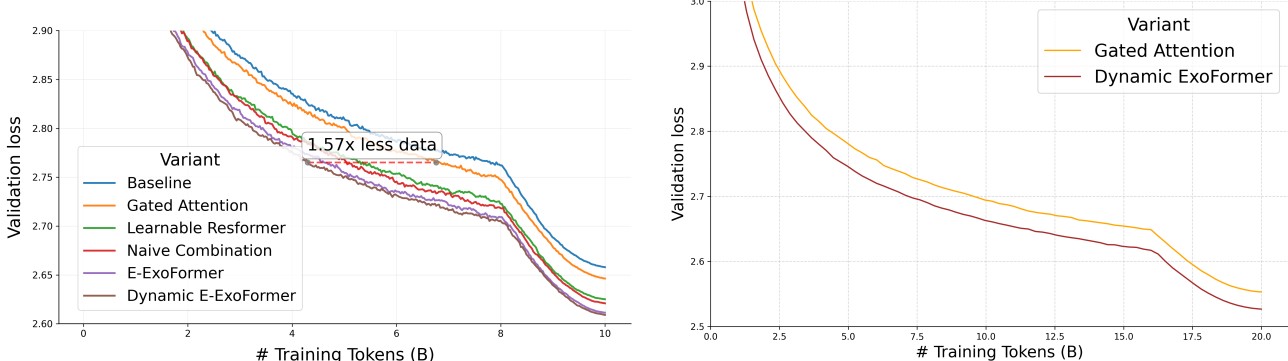

*Figure 2.* (Left) Validation loss for $\sim 450M$ parameter models. (Right) Validation loss for $\sim 1B$ parameter models.

Our primary contribution is **ExoFormer**, which resolves this tension by learning a set of *dedicated exogenous anchor projections* outside the sequential layer stack. The internal-anchor version, which we term NuResFormer (**N**ormalized **u**nified), serves as a comparative foundation. Decoupling of the two roles proves consistently beneficial: every Exo-Former variant outperforms its corresponding NuResFormer counterpart in perplexity while remaining competitive in downstream accuracy despite no increase in width or depth. We explain this high performance via an *Offloading Hypothesis*: the external anchor assumes the role of preserving token identity, allowing the sequential layers to specialize almost exclusively in refinement. Additionally, we show that anchoring on slightly deeper representations can further improve perplexity while increasing throughput.

## 2. Related Work

**Value Residuals and Gated Attention.** Zhou et al. (2025) introduced *Value Residual Learning* (ResFormer), which adds a residual connection from the first layer's value projection ($V_1$) to the value projections of all subsequent layers (regulated by scalar coefficients $\lambda_1$ and $\lambda_2$). This simple yet effective method was shown to greatly improve model performance and data efficiency, highlighting the benefit of explicitly reusing early content representations. However, preliminary attempts to residualize queries and keys were found to be unstable (Zhou et al., 2025). Concurrently, gated attention mechanisms have been explored to introduce dynamic, input-dependent modulation to the attention output, improving expressiveness and training stability (Qiu et al., 2025). Our work generalizes residual learning to all attention pathways and stabilizes it via normalized mixing, bridging the gap between residual connections and gated attention.

**Cross-layer communication and residual mixing.** Recent work has sought to improve cross-layer information flow in Transformers beyond simple residual connections.

Zhu et al. (2025) proposed Hyper-Connections that expand residual stream width. More recently, Xie et al. (2026) introduced Manifold-Constrained Hyper-Connections (mHC), restoring an identity-like signal-preservation property to the hyper-connected architecture.

Most closely related to our work, though orthogonal in direction, is **MUDDFormer** (Xiao et al., 2025), which proposes Multiway Dynamic Dense (MUDD) connections. MUD-DFormer decouples the input to each Transformer block into four streams (query, key, value, residual) and dynamically aggregates outputs from *all* preceding layers using context-specific weights generated by a small MLP.

While the techniques mentioned above and our work focus on enhancing residual pathways, they operate at different levels and can coexist. MUDDFormer primarily addresses the *input* to the attention projections. In contrast, our unified mixing framework focuses on *mixing* the attention projections themselves. Our approach can be seen as a form of blending *after* projection, whereas MUDDFormer enriches the input *before* projection.

**First-layer reuse, KV efficiency, and low-rank cross-layer structures.** Wu et al. (2025) proposed SkipV1Former, which substitutes half of each deeper layer's Value heads with first-layer heads, cutting KV cache by $\sim 25\%$ while improving perplexity. Sun et al. (2024) proposed YOCO, which caches global KV pairs once and reuses them via cross-attention; while effective for inference-time memory reduction, it relies on static sharing rather than learned mixing. From a parameter-efficiency perspective, Kong et al. (2025) discovered that inter-layer activation residuals are low-rank and leveraged this via cross-layer low-rank differences to reduce trainable parameters. These approaches are complementary but distinct from ours: ExoFormer resolves the first-layer tension by *decoupling* the anchor from the sequential stack, operating in the full-rank projection space, and enabling learnable, normalized mixing across *all* attention pathways.

# 3. Methodology

## 3.1. Preliminaries and Notation

For the $n$-th Transformer layer, let $H_{n-1} \in \mathbb{R}^{T \times d_{\text{model}}}$ be the input hidden states (after pre-normalization), where $T$ is the sequence length and $d_{\text{model}}$ is the model width. We use $h$ attention heads and per-head dimension $d_k$ such that $d_{\text{model}} = h\, d_k$. We denote the projected queries, keys, values, and gate logits as $Q_n, K_n, V_n, G_n$. Let $\text{RMSNorm}(\cdot)$ denote per-token RMS normalization applied independently to each attention head (with learnable gain).

We consider three coefficient granularities for $\lambda$: Scalar (S) (a single coefficient shared across all channels), Headwise (H) (one coefficient per head, broadcast across $d_k$), and Elementwise (E) (one coefficient per channel). Elementwise and headwise granularities grant fine-grained control over reuse mechanisms. We further analyze how they impact training and downstream performance.

## 3.2. Multi-Head Attention

We describe attention as a sequence of stages, with two standard enhancements integrated implicitly: rotary position embeddings (RoPE) (Su et al., 2024) and query/key normalization (QKNorm) (Henry et al., 2020). For simplicity, the multi-head mechanism is presented using unified tensors with an implicit head dimension.

**Stage 1: QKV Linear Projections.** Given $H_{n-1} \in \mathbb{R}^{T \times d_{\text{model}}}$, we compute the projected tensors:

$$Q_n = H_{n-1}W_n^Q, \quad K_n = H_{n-1}W_n^K, \quad V_n = H_{n-1}W_n^V, \tag{1}$$

where $W_n^Q, W_n^K, W_n^V \in \mathbb{R}^{d_{\text{model}} \times d_{\text{model}}}$. The resulting $Q_n, K_n, V_n$ are structured to contain $h$ heads implicitly.

**Stage 2: Scaled Dot-Product Attention (SDPA).** Attention is computed per head, which is represented here as a single operation:

$$
\begin{aligned}
A_n &= \text{softmax}\left(\frac{Q_n K_n^\top}{\sqrt{d_k}}\right) \in \mathbb{R}^{T \times T}, \\
U_n &= A_n V_n \in \mathbb{R}^{T \times d_{\text{model}}},
\end{aligned}
\tag{2}
$$

where the operations encompass the independent computations across $h$ heads.

**Stage 3: Final Output Projection.** The output of the attention computation is projected:

$$O_n = U_n W_n^O, \qquad W_n^O \in \mathbb{R}^{d_{\text{model}} \times d_{\text{model}}}. \tag{3}$$

In NuResFormer/ExoFormer, the above stages use mixed tensors $\widehat{Q}_n, \widehat{K}_n, \widehat{V}_n$ from Eq. (10).

## 3.3. Gated Attention

We adopt the elementwise head-specific multiplicative gating formulation as described by Qiu et al. (2025). The general form is:

$$Y' = g(Y, X, W_\theta, \sigma) = Y \odot \sigma(X W_\theta), \tag{4}$$

where $Y$ is the feature tensor to be modulated, $X$ is the input used to compute the gating scores, $W_\theta$ are learnable parameters, $\sigma$ is an activation function (sigmoid unless stated otherwise), and $\odot$ denotes elementwise multiplication.

In our attention block, we apply gating to the concatenated multi-head output (i.e., after Stage 2 and before Stage 3). Let $G_n \in \mathbb{R}^{T \times (h d_k)}$ denote the gate logits, computed from $H_{n-1}$:

$$G_n = H_{n-1}W_n^G, \qquad W_n^G \in \mathbb{R}^{d_{\text{model}} \times d_{\text{model}}}. \tag{5}$$

Then the gated attention output is

$$\widetilde{U}_n = U_n \odot \sigma(G_n), \qquad O_n = \widetilde{U}_n W_n^O. \tag{6}$$

**Normalization placement.** We note that in ablations, using post-norm instead of pre-norm substantially degraded gated attention performance.

## 3.4. Unified Mixing Formulation

We now present a general framework for mixing. The core idea is to enrich each layer's attention pathways with a set of persistent *anchor projections* $\{Q_{\text{anc}}, K_{\text{anc}}, V_{\text{anc}}, G_{\text{anc}}\}$ that is reused, via learnable mixing, across all layers.

**Current-layer projections.** For layer $n$ with input $H_{n-1}$, we compute the standard projections:

$$
\begin{aligned}
Q_n &= H_{n-1}W_n^Q, \quad K_n = H_{n-1}W_n^K, \\
V_n &= H_{n-1}W_n^V, \quad G_n = H_{n-1}W_n^G.
\end{aligned}
\tag{7}
$$

**Anchor projections.** The anchors are a fixed set of projections defined once for the entire model. In our work, we explore two instantiations:

- **NuResFormer:** The anchors are the projections from the very first attention layer. That is,

$$Q_{\text{anc}} = Q_1, \; K_{\text{anc}} = K_1, \; V_{\text{anc}} = V_1, \; G_{\text{anc}} = G_1, \tag{8}$$

- **ExoFormer:** The anchors are produced by a dedicated, external projection module on the input embeddings:

$$
\begin{aligned}
Q_{\text{anc}} &= H_0 W_{\text{anc}}^Q, \quad K_{\text{anc}} = H_0 W_{\text{anc}}^K, \\
V_{\text{anc}} &= H_0 W_{\text{anc}}^V, \quad G_{\text{anc}} = H_0 W_{\text{anc}}^G,
\end{aligned}
\tag{9}
$$

where $W_{\text{anc}}^Q, W_{\text{anc}}^K, W_{\text{anc}}^V, W_{\text{anc}}^G \in \mathbb{R}^{d_{\text{model}} \times d_{\text{model}}}$ are independent learnable weight matrices.

**Mixing with normalized sources.** For each component $S \in \{Q, K, V, G\}$, we mix the anchor projection with the current-layer projection using learned coefficients. To stabilize the mixture, we apply RMS normalization to the anchor source before scaling:

$$\widehat{S}_n = \lambda_{n,1}^S \odot \text{RMSNorm}(S_{\text{anc}}) + \lambda_{n,2}^S \odot S_n, \\ \forall S \in \{Q, K, V, G\}, \tag{10}$$

where $\lambda_{n,1}^S, \lambda_{n,2}^S$ are coefficient tensors of the chosen granularity (scalar, headwise, or elementwise). All $\lambda$ parameters are initialized to 0.5 and are learnable.

We apply QKNorm and RoPE to the mixed queries and keys, $\widehat{Q}_n$ and $\widehat{K}_n$. We then compute the scaled dot-product attention per head using the mixed projections $\widehat{Q}_n, \widehat{K}_n, \widehat{V}_n$, apply the gating operation using $\widehat{G}_n$, and finally pass the result to the output projection.

### 3.5. First Layer Tension and ExoFormer

Cross-layer residual reuse makes early information available at every depth. This is powerful, but it implicitly forces the first layer to satisfy two pressures:

1. **Reusable anchor:** produce a broadly useful reference representation that remains valuable throughout depth.

2. **Progressive computation:** produce features that are easy for downstream layers to transform into increasingly task-relevant abstractions.

While these roles appear misaligned (universal anchors favor invariance, while progressive computation necessitates change), they can theoretically coexist because it is an *optional pathway* modulated by learned mixing coefficients $(\lambda_{n,1}, \lambda_{n,2})$. However, this structural constraint places pressure on the first layer to make compromises, inherently limiting its effectiveness in both roles. This tension is empirically supported by the analysis in Appendix Figure 8, showing that NuResFormer's first layer adopts a permissive gating policy compared to standard baselines, indicating a compromise on selectivity in favor of serving as a stable anchor.

Our most successful approach, termed **ExoFormer**, instantiates the general framework of Section 3.4 with the dedicated exogenous projections of Eq. (9).

### 3.6. Dynamic Mixing (DM) Module

Building upon the unified formulation, we take inspiration from MUDDFormer's Depth-wise Aggregate (DA) module (Xiao et al., 2025) and introduce a dynamic variant where the learnable parameters are modulated by context-dependent scaling factors computed from the layer input

$H_{n-1}$ using a small MLP. This allows the model to adapt its mixing strategy based on the specific context.

**Dynamic Coefficient Generation.** For each layer $n$, we compute modulation scalars from its input $H_{n-1}$ (pre-normalized) using a two-layer MLP with GELU activation and sigmoid output:

$$\mathcal{DM}_n(H_{n-1}) = \sigma\Big(\text{GELU}\big(H_{n-1}W_{n,1}^{\text{DM}}\big)W_{n,2}^{\text{DM}} + b_n^{\text{DM}}\Big) \tag{11}$$

The trainable parameters for the Dynamic Mixing module at layer $n$ are:

$$\theta_n^{\text{DM}} = \big\{ W_{n,1}^{\text{DM}} \in \mathbb{R}^{d_{\text{model}} \times 16}, \ W_{n,2}^{\text{DM}} \in \mathbb{R}^{16 \times 8}, \ b_n^{\text{DM}} \in \mathbb{R}^8 \big\}$$

The output dimension of this module is 8, corresponding to the dynamic scaling factors:

$$\{\gamma_{n,1}^Q, \gamma_{n,2}^Q, \gamma_{n,1}^K, \gamma_{n,2}^K, \gamma_{n,1}^V, \gamma_{n,2}^V, \gamma_{n,1}^G, \gamma_{n,2}^G\}$$

The output layer weights $W_{n,2}^{\text{DM}}$ and bias $b_n^{\text{DM}}$ are zero-initialized, ensuring that initial sigmoid outputs are 0.5. Consequently, the base $\lambda$ parameters must be initialized at 1.0 to achieve effective identity mixing at initialization.

**Modulated Mixing.** For each component we compute using the dynamic scaling factors:

$$\widehat{S}_n = (\lambda_{n,1}^S \gamma_{n,1}^S) \odot \text{RMSNorm}(S_{\text{anc}}) + (\lambda_{n,2}^S \gamma_{n,2}^S) \odot S_n, \\ \forall S \in \{Q, K, V, G\}, \tag{12}$$

where $\gamma_{n,i}^S$ are broadcast appropriately based on the residual granularity (elementwise, headwise, or scalar), and $\lambda_{n,i}^S$ are the learnable base parameters.

## 4. Experiments

### 4.1. Experimental Setup

**Training Details** All models use a modern pre-normalized Transformer architecture with SwiGLU activations (Shazeer, 2020), QKNorm (Henry et al., 2020), and rotary position embeddings (Su et al., 2024). We follow prior work by initializing projection and classification layers to zero (Yang et al., 2022), removing bias terms (except for the dynamic mixing module) (Chowdhery et al., 2023), applying z-loss regularization (de Brébisson & Vincent, 2016), and disabling dropout. Training is performed on the FineWeb-Edu dataset (Penedo et al., 2024). Specifically, the $\sim 450M$ parameter models are trained on 10B tokens, while the $\sim 1B$ parameter models are trained on 20B tokens.

*Table 1.* Performance comparison of model variants on 6 multiple-choice downstream tasks. The parameter counts are: 453M for models with gated attention, 454M for those without it, and 457M for models that incorporate an external anchor. Naïve Combination refers to the unmodified addition of Gated Attention and ResFormer. Dynamic ExoFormer achieves the highest overall accuracy and the lowest perplexity, demonstrating the advantages of decoupling the anchor from the first layer and using context-aware mixing. All models use full normalization on anchors unless noted otherwise. The notation {V,G} denotes ablations where only value and gate projections are mixed; queries and keys are computed solely from the current layer. Prefixes E, H, and S denote elementwise, headwise, and scalar mixing granularity. "Only Q/K Norms" applies RMSNorm solely to anchor queries/keys, while "No Norm" omits anchor normalization.

| MODEL | ARC-C | ARC-E | HELLA. | OBQA | PIQA | WINO. | AVG. ACC | PPL |
|---|---|---|---|---|---|---|---|---|
| **BASELINES** | | | | | | | | |
| BASE TRANSFORMER | 30.97 | 63.38 | 42.02 | 33.60 | 67.30 | 51.54 | 48.14 | 14.79 |
| GATED ATTENTION | 30.89 | 64.69 | 42.73 | 34.00 | 67.14 | 53.35 | 48.80 | 14.64 |
| RESFORMER (VALUE RESIDUAL) | 33.62 | 64.86 | 43.49 | 34.40 | **68.88** | 52.64 | 49.65 | 14.32 |
| NAÏVE COMBINATION | 32.94 | 64.52 | 43.47 | 32.60 | 68.34 | 52.64 | 49.09 | 14.25 |
| **INTERNAL ANCHOR** | | | | | | | | |
| E-NURESFORMER (ONLY Q/K NORMS) | 31.74 | 64.27 | 43.83 | 33.00 | 68.23 | 52.41 | 48.91 | 14.21 |
| H-NURESFORMER (ONLY Q/K NORMS) | **34.73** | 64.06 | 44.45 | 34.20 | 67.57 | 53.04 | 49.68 | 14.21 |
| S-NURESFORMER (ONLY Q/K NORMS) | 32.85 | **66.08** | 43.87 | 34.20 | 68.28 | 53.67 | 49.83 | 14.22 |
| E-NURESFORMER (NO NORM, {V,G}) | 32.42 | 64.94 | 43.80 | 35.00 | 67.52 | 51.54 | 49.20 | 14.24 |
| H-NURESFORMER (NO NORM, {V,G}) | 31.23 | 64.18 | 43.50 | 34.00 | 67.68 | 54.06 | 49.11 | 14.22 |
| S-NURESFORMER (NO NORM, {V,G}) | 31.83 | 64.44 | 43.66 | 34.80 | 68.44 | 52.49 | 49.28 | 14.24 |
| E-NURESFORMER | 33.70 | 65.07 | 44.23 | 33.60 | 68.34 | 53.12 | 49.68 | 14.15 |
| H-NURESFORMER | 33.79 | 65.11 | 44.09 | 33.40 | 67.41 | 52.72 | 49.42 | 14.17 |
| S-NURESFORMER | 32.42 | 64.02 | 43.93 | 33.20 | 67.90 | 53.20 | 49.11 | 14.17 |
| **EXTERNAL ANCHOR** | | | | | | | | |
| E-EXOFORMER (NO NORM) | 32.08 | 63.76 | 43.49 | 34.40 | 68.77 | **55.64** | 49.69 | 14.30 |
| **DYNAMIC E-EXOFORMER** | 33.36 | 65.87 | **44.54** | 34.40 | 68.28 | 55.17 | **50.27** | **14.09** |
| E-EXOFORMER | 32.42 | 64.65 | 44.28 | **36.40** | 67.74 | 53.59 | 49.85 | 14.13 |
| H-EXOFORMER | 32.17 | 65.87 | 44.00 | 32.40 | 68.23 | 52.72 | 49.23 | 14.14 |
| S-EXOFORMER | 32.17 | 65.66 | 44.33 | 33.60 | 68.06 | 51.14 | 49.16 | 14.15 |

We optimize matrix parameters with Muon (Polar Express variant (Amsel et al., 2025)), applying a cautious weight decay of 0.1 (Chen et al., 2025), while 1D parameters are trained using Adam. We selected Muon as a state-of-the-art optimizer for large-scale LLM training (Liu et al., 2025), though we anticipate that the observed improvements should be optimizer-agnostic. Gradient norms are clipped to 1.0, and all models follow the same optimization setup, to ensure fair comparison across architectures. Training uses a global batch size of 262,144 tokens and a sequence length of 2,048.

Full hyperparameters are provided in the Appendix. All experiments are conducted on a single NVIDIA H100 80GB GPU using native BF16 precision, with FlashAttention (Dao et al., 2022) enabled.

**Evaluation Details**    We evaluate each benchmark example using a 5-shot prompt. To reduce length-related bias, we report length-normalized accuracy whenever possible. Perplexity is measured on the FineWeb-Edu validation set containing 100 million tokens. We report results on 6 multiple-choice benchmarks: ARC_CHALLENGE, ARC_EASY (Clark et al., 2018), HELLASWAG (Zellers et al., 2019), OPEN-BOOKQA (Mihaylov et al., 2018), PIQA (Bisk et al., 2020),

and WINOGRANDE (Sakaguchi et al., 2020).

### 4.2. The Role of Anchor Normalization

RMSNorm serves two roles: first, as a scale-dependent rational function, it introduces a non-linear transformation into the otherwise linear residual pathway; and second, as a mild isotropization operator, RMSNorm independently normalizes each token representation across the heads, rescaling the vector to unit RMS while preserving directional information and removing per-token scale differences. This per-token operation retains the relative alignment between dimensions, which encodes semantic and syntactic features.

The necessity of applying RMSNorm to anchor sources is empirically supported by an analysis of the learned mixing coefficients in unnormalized models. For instance, in the ExoFormer variant without residual normalization, the proportion of near-zero coefficients (below 0.001) for $\lambda_1$ (the strength of anchor signal) is approximately triple that of the normalized variant. This suppression of the anchor pathway suggests the model is actively compensating for distributional mismatch.

We hypothesize that the naïve combination underperforms

because unnormalized mixing introduces a distributional mismatch between the anchor and current-layer projections, forcing the gating mechanism to divert capacity from selective filtering toward compensatory scale correction. Consequently, $\sigma(G)$ must compensate for this mismatch rather than performing its intended filtering role, resulting in instability during training (Figure 2) and worse downstream accuracy than ResFormer in isolation. This instability is quantitatively reflected in gradient diagnostics: the naïve combination exhibits a gradient spike score of 0.0108, higher than Gated Attention (0.0085) and two times higher than E-ExoFormer (0.0054). The full breakdown across all model variants is provided in Appendix F.

### 4.3. Extending Mixing to Q, K, and G Pathways

**The Instability of Unnormalized Q/K Residuals and the Stabilizing Role of QKNorm.** Consistent with prior observations (Zhou et al., 2025), we find a clear hierarchy of stability. Models attempting to use unnormalized Q/K residuals without QKNorm proved difficult to optimize, exhibiting divergent loss in preliminary runs. Critically, adding QKNorm alone stabilizes training and recovers baseline performance. We hypothesize this is because QKNorm compresses the scale of $Q$ and $K$, mitigating the distributional mismatch that arises when injecting early-layer routing signals into deeper layers. Furthermore, adding explicit RMSNorm to the Q/K anchor sources on top of QKNorm yields the best results, enabling positive reuse.

**Gating Logit Mixing Is Inherently More Stable.** In contrast to $Q$ and $K$, residual connections for the gating logits $G$ were straightforward to incorporate even without normalization. We hypothesize that gate logits, which are compressed by the sigmoid function, are naturally less sensitive to distributional mismatch.

**Granularity in NuResFormer.** Among NuResFormer configurations, scalar mixing achieves the highest average downstream accuracy (49.83%) despite having the fewest parameters. Headwise mixing performs nearly as well (49.68% to 49.42%), indicating that allocating one degree of freedom per attention head captures a significant portion of the beneficial structure. Elementwise mixing yields the best language modeling perplexity (14.15 under full residual norms) but slightly lower downstream accuracy than its scalar counterpart. This suggests increased parametric freedom can improve in-distribution loss but may not generalize as well.

**Granularity in ExoFormer.** ExoFormer exhibits a notably different profile. Here, elementwise mixing (E-ExoFormer) achieves the overall best performance, attaining the highest average accuracy (49.85%) and the lowest validation perplexity (14.13) of any static model. This reversal

indicates that the effectiveness of coefficient granularity is architecture-dependent. We hypothesize that in this cleaner setting (due to decoupling), the optimizer can effectively exploit fine-grained mixing to orchestrate precise reuse policies without compromising generalization.

**Emergent Head Structure in Elementwise Mixing.** Visualizing the learned elementwise coefficients reveals emergent patterns specific to each attention component (Figure 3). For the value pathway ($V$), the heatmaps exhibit *sharp boundaries that align precisely with head blocks*. This structure emerges despite the optimizer having the freedom to set each channel independently, aligning with previous research on heads as specialized submodules for routing information (Voita et al., 2019).

Conversely, queries ($Q$), keys ($K$), and gate logits ($G$) exhibit *finer-grained, intra-head structure*, such as alternating bands of high and low reuse ("striping"), suggesting a form of sub-head specialization. Notably, the positions of these high/low bands in $Q$ often align with those in $K$.

**Flexibility Enabled by Dynamic Mixing.** As shown in Figure 3a, $\lambda_{n,1}/\lambda_{n,2}$ for queries, keys, and gating logits are substantially more uniformly distributed across channels and layers in the dynamic variant. Rather than converging to a fixed, layer-specific reuse policy, the model can modulate the strength of the anchor pathway for each component in real time based on the input sequence, enabling more freedom in expression in the mixing coefficients themselves.

### 4.4. Mix-Compress-Refine Theory, Gated Attention, and ResFormer

As shown in Figure 4, the baseline Transformer's behavior aligns with the Mix-Compress-Refine theory proposed by Queipo-de-Llano et al. (2025): (1) it begins with a high attention entropy phase for broad, contextual integration of token information, (2) transitions into a compression valley marked by dominant attention sinks and a drastic drop in core features that halt mixing and reduce representational dimensionality to filter out useless information, (3) concludes with a sudden rise in attention entropy as sinks dissipate, enabling the refined, token-specific processing necessary for generation.

While these three stages emerge naturally in standard Transformers, they are not optimized for computational efficiency. The model expends computational capacity collapsing contextual information, only to subsequently reconstruct it, indicating inefficiency in the standard architecture (Figure 4c).

**Gated Attention and residual mixing improve performance by targeting stage 2.** Based on the empirical patterns observed, we propose that the performance gains from

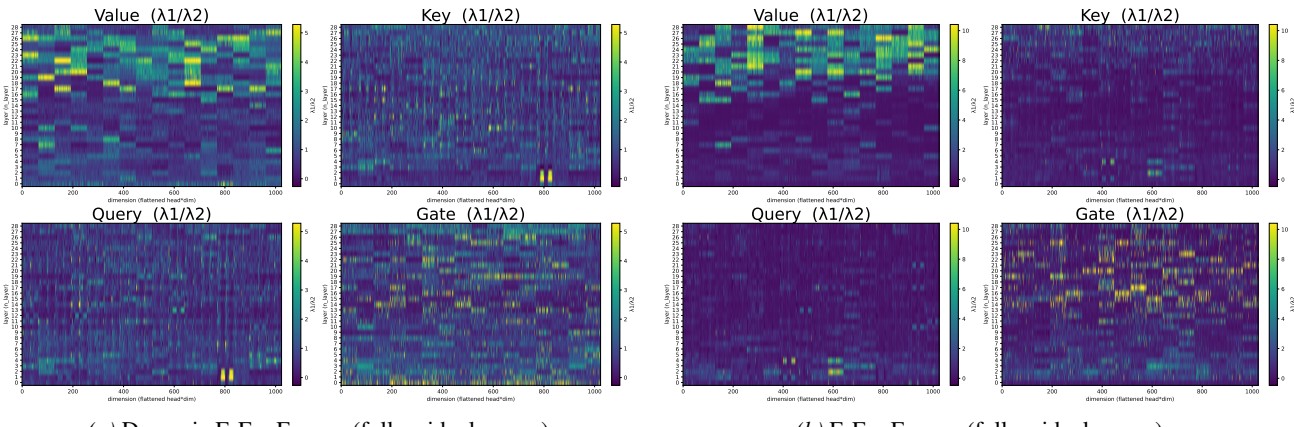

*(a)* Dynamic E-ExoFormer (full residual norms)                 *(b)* E-ExoFormer (full residual norms)

*Figure 3.* Heatmaps showing the learned mixing coefficient ratio $\lambda_{n,1}/\lambda_{n,2}$ for each residualized component $\{Q, K, V, G\}$ across layers (y-axis) and channels/heads (x-axis) for model variants $\sim 450M$. This ratio quantifies the model's reliance on the anchor relative to the current layer's projection; a higher value indicates stronger reuse of the early signal.

gated attention and residual mixing may partly stem from their effect on the model's second, compression stage.

The gating mechanism addresses the model's need to filter irrelevant context, a core function of stage 2. By introducing input-dependent sparsity that selectively modulates information flow at every layer, gated attention provides a form of distributed filtering. This may reduce or eliminate the need for a sharp, dedicated compression phase characterized by dominant attention sinks. The empirical evidence is consistent with this interpretation: as shown in Figure 4c and Table 2, models with gated attention exhibit no clear compression valley and show a drastic reduction in attention sink magnitude compared to the baseline. This suggests the gates could act as a learned, adaptive replacement for the more rigid, sink-based filtering.

Similarly, in models employing residual mixing, the residual pathways may provide an implicit, learned alternative to abrupt sink-based filtering. The learned blending could reduce the model's reliance on extreme, attention-sink-driven compression to isolate useful signals. This is supported by the milder compression valley observed in such models (Figure 4c) and their notable reduction in attention sink magnitude (Table 2).

These findings suggest that both mechanisms alleviate the abrupt information bottleneck characteristic of stage 2, replacing inefficient, sink-based filtering with a more continuous and learned process for isolating relevant information.

### 4.5. The Offloading Hypothesis: Specialization via Exogenous Anchors

The architectural decoupling in ExoFormer enables an interesting functional specialization, which we formalize as the Offloading Hypothesis. By providing a dedicated, high-

fidelity source of token identity, the exogenous anchor allows the sequential layers to offload the burden of preserving static features and specialize almost exclusively in the final "refinement" stage (Stage 3) of the Mix-Compress-Refine cycle.

Evidence for this specialization is clearly visible in the token-similarity trajectory (Figure 4b). All models exhibit a characteristic pattern: similarity rises to an early maximum (Stage 1: mixing), dips to a local minimum (Stage 2: compression), rises again, and finally falls in the last layer (Stage 3: refinement). We interpret the local minimum as the onset of Stage 3. Crucially, ExoFormer variants and NuResFormer spend approximately two-thirds of their layers in this specialized refinement stage, the highest proportion of any model, while the baseline spends only one-third.

This specialization is possible because exogenous anchors fundamentally reconfigure information management within the Transformer. In a standard architecture, the residual stream must concurrently carry two conflicting types of information: (1) distinct, static features that preserve token identity ("What token am I?"), and (2) transformed, task-relevant features that evolve through the layers to support next-token prediction. Over-smoothing in standard models catastrophically loses the first type, crippling the model's ability to route information effectively. ExoFormer circumvents this tension by introducing a persistent external anchor that is reinjected at every layer, guaranteeing access to high-fidelity token identity.

This architecture enables a form of **functional offloading** that separates these concerns:

1. **The Anchor Specializes in "Identity Preservation."** By providing a guaranteed, high-fidelity source of token distinctiveness at every layer, the exogenous an-

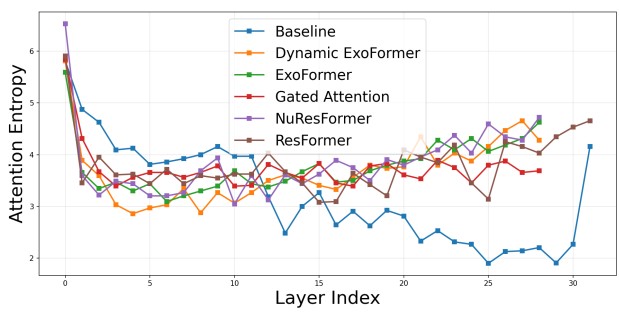

*(a)* Attention entropy by layer.

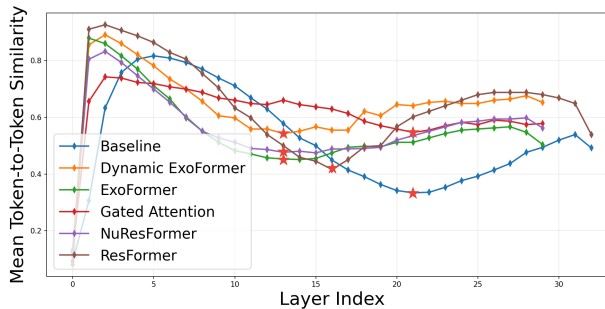

*(b)* Average cosine similarity between different token representations within the same layer.

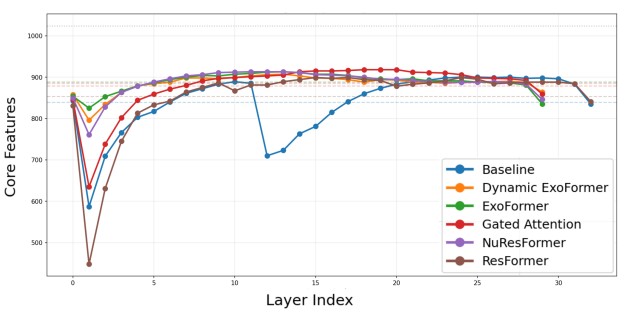

*(c)* PCA Core Features by layer to explain 99% variance.

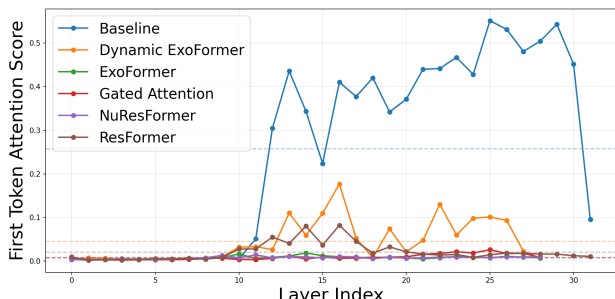

*(d)* First-token attention (attention sink) by layer.

*Figure 4.* Attention-pattern and representation analysis across model variants $\sim 450M$. *Elementwise* is used unless stated otherwise. Some graphs include input embeddings for comparison. The stars on the cosine similarity diagram represent the onset of refinement.

*Table 2.* Metrics averaged across layers for $\sim 450M$ models.

| MODEL | ATTENTION SINK | TOKEN SIMILARITY | PCA CORE FEATURES |
|---|---|---|---|
| BASELINE | 0.2580 | **0.5364** | 839.0 |
| DYNAMIC EXOFORMER | 0.0453 | 0.6606 | 885.0 |
| EXOFORMER | 0.0078 | 0.5682 | **890.1** |
| GATED | 0.0091 | 0.6327 | 880.0 |
| NURESFORMER | **0.0077** | 0.5785 | 887.9 |
| RESFORMER | 0.0212 | 0.6558 | 853.8 |

chor removes the requirement for the main residual stream to preserve static token features. The dedicated anchor parameters ($W_{\text{anc}}^{Q}, W_{\text{anc}}^{K}, W_{\text{anc}}^{V}, W_{\text{anc}}^{G}$) can specialize exclusively in maximizing variance and distinctiveness, optimized to serve as a universal reference.

2. **The Sequential Layers Specialize in "Progressive Computation."** With identity information externally supplied, the internal layers are decoupled from the constraint of preserving early-layer geometry. They are no longer required to maintain a stable, high-fidelity record of token identity within the residual stream, a task that inherently conflicts with the need to transform and abstract features.

Unburdened by the need to preserve static token features, the layers can more efficiently allocate their capacity to

learning complex, high-level patterns, leading to the observed improvements in both perplexity and downstream accuracy. The architecture effectively separates the stable, memory-like function of identity preservation (handled by the exogenous anchor) from the adaptive, computational function of progressive refinement (handled by the sequential layers), resulting in a more efficient and performant model.

We strengthen the causal interpretation of the Offloading Hypothesis through two controlled interventions beyond the correlational analyses above. First, we perform the anchor-removal ablation on *both* architectures. When the anchor contribution is zeroed out ($\lambda_{n,1} = 0$) during inference, NuResFormer suffers an even more catastrophic collapse than ExoFormer: its final-layer PCA core features plummet to only 181 dimensions—far below ExoFormer's

321—while token-to-token similarity peaks even higher. This asymmetry directly supports the First-Layer Tension hypothesis (Section 3.5): because NuResFormer forces the first layer to serve as both a computational block and a reusable anchor, the resulting internal anchor is fundamentally more brittle when removed.

Second, we conduct the inverse ablation by zeroing out the current-layer contribution ($\lambda_{n,2} = 0$), forcing the model to rely exclusively on the anchor. ExoFormer retains 822 PCA core features averaged across layers, whereas NuResFormer retains only 611. This confirms that the exogenous anchor provides a significantly higher-fidelity identity signal than the compromised internal anchor.

Full methodology, graphs, and additional metrics are provided in Appendix C.

### 4.6. Anchor Depth: $H_0$, $H_1$, and $H_2$

Because the exogenous anchor is decoupled from the sequential layer stack, its source representation need not be restricted to the input embeddings $H_0$. We investigate whether anchoring on slightly deeper representations, $H_1$ or $H_2$, improves performance.

Table 3 reports validation perplexity for the $\sim$450M models after 10B tokens. Surprisingly, anchoring on $H_1$ yields the best result (PPL 14.02), outperforming both the $H_0$ anchor (14.09) and the $H_2$ anchor (14.09). A static ExoFormer using an $H_1$ anchor also outperforms the dynamic $H_0$ baseline (14.07 vs. 14.09).

*Table 3.* Validation perplexity of E-ExoFormer variants with different anchor depths ($\sim$450M, 10B tokens).

| MODEL | ANCHOR SOURCE | VAL. PPL |
|---|---|---|
| DYNAMIC | $H_0$ (INPUT EMBEDDINGS) | 14.09 |
| DYNAMIC | $H_1$ (LAYER 1) | **14.02** |
| DYNAMIC | $H_2$ (LAYER 2) | 14.09 |
| STATIC | $H_0$ (INPUT EMBEDDINGS) | 14.13 |
| STATIC | $H_1$ (LAYER 1) | 14.07 |

This result contrasts with Value Residual Learning (ResFormer), which finds the earliest representation ($H_0$) optimal (Zhou et al., 2025). We attribute the difference to the *exogenous* nature of our anchor: because the anchor module is decoupled from the sequential layer stack, it can effectively leverage the richer semantic signal from $H_1$ without suffering the first-layer tension inherent to internal-anchor designs. We hypothesize that $H_1$ provides an optimal "effective anchor depth". It preserves core token identity while offering slightly more processed features. Anchoring on $H_2$ does not improve further, suggesting a saturation point beyond which the anchor loses the high-fidelity identity signal required by the offloading mechanism. In addition to

the perplexity gains, $H_1$ and $H_2$ anchors improve throughput compared with the $H_0$ baseline by reducing the mixing overhead.

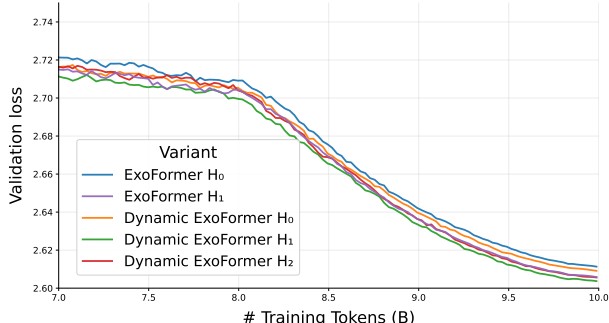

*Figure 5.* Training curves (validation perplexity) for Dynamic ExoFormer with anchor sources $H_0$, $H_1$, and $H_2$. The $H_1$ variant converges faster and to a lower final loss.

## 5. Conclusion

We introduced ExoFormer, a novel Transformer architecture that decouples token identity preservation from computational refinement by learning dedicated exogenous anchor projections. Through a unified normalized mixing framework across queries, keys, values, and gates, ExoFormer resolves the architectural tension inherent in reusing first-layer projections. Experiments demonstrate that this decoupling consistently improves performance, with the dynamic variant achieving significant gains in downstream accuracy and data efficiency compared to standard gated attention baselines. These results validate the Offloading Hypothesis, suggesting that externalizing identity preservation enables sequential layers to specialize exclusively in high-level feature transformation, providing an efficient architectural path for enhancing large language models.

## Impact Statement

The primary positive societal impact of this work is environmental sustainability. By reducing the number of tokens required to train high-performing models, ExoFormer directly addresses the escalating energy consumption and carbon footprint associated with training large language models (LLMs). The open-source release of code and pre-trained models promotes democratization, lowering the computational barriers for researchers and smaller organizations to develop and deploy state-of-the-art NLP tools. Furthermore, by providing a theoretical framework for how information flows through deep networks (the Offloading Hypothesis), this work aids in the interpretability and mechanistic understanding of LLMs, which is crucial for diagnosing failure modes and ensuring model reliability.

While our contribution is architectural (focused on efficiency

and structure rather than specific content generation), it facilitates the deployment of more capable AI systems that could be employed maliciously. Additionally, while our ablations suggest robustness, there is inherent uncertainty in how the "offloading" mechanism will scale to significantly larger parameter counts (e.g., 100B+), which may introduce novel failure modes or safety challenges not observed in the 450M–1B range studied here.

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

## A. Anchor Reliance and Hidden State Similarity

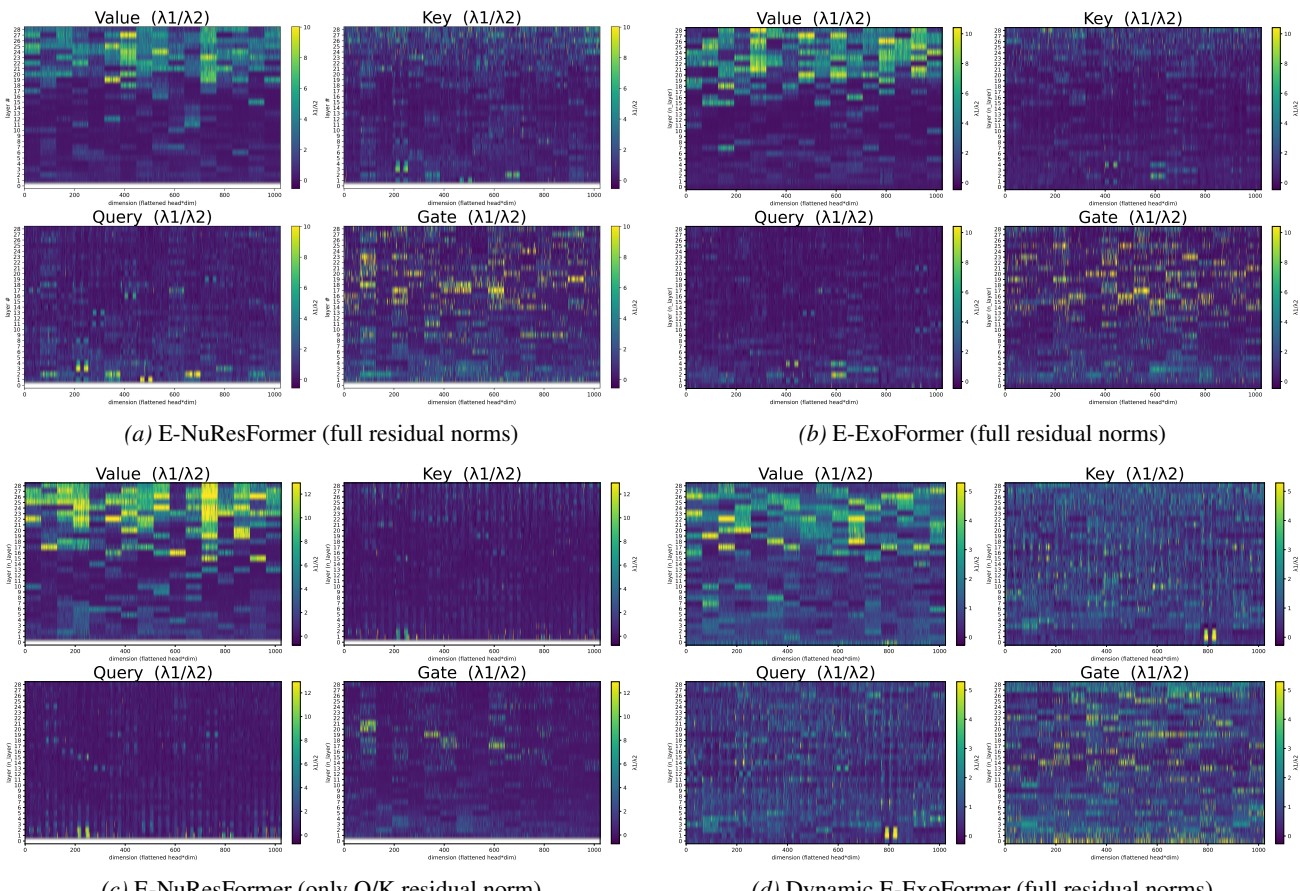

*(a)* E-NuResFormer (full residual norms)  *(b)* E-ExoFormer (full residual norms)

*(c)* E-NuResFormer (only Q/K residual norm)  *(d)* Dynamic E-ExoFormer (full residual norms)

*Figure 6.* Heatmaps showing the learned mixing coefficient ratio $\lambda_{n,1}/\lambda_{n,2}$ for each residualized component $\{Q, K, V, G\}$ across layers (y-axis) and channels/heads (x-axis) for model variants $\sim 450M$. This ratio quantifies the model's reliance on the anchor relative to the current layer's projection; a higher value indicates stronger reuse of the early signal. Even with elementwise freedom, the learned pattern is largely constant within heads for $V$, while $Q$, $K$, and $G$ exhibit finer-grained structure. The dynamic variant shows more uniform distributions across channels and layers.

## B. Gate Activation Profiles and First-Layer Selectivity

NuResFormer's first layer exhibits **a high mean gate activation** (approximately 0.4-0.5), indicating its gating mechanism is not highly suppressive, allowing roughly half of the attention output to pass through. This contrasts sharply with standalone Gated Attention, where the first-layer mean activation is significantly lower (approximately 0.2). Following this initial peak, gate activations fall rapidly in intermediate layers before rising steadily again in deeper layers.

This pattern provides direct empirical support for the architectural tension hypothesized in Section 3.5. When the first layer also serves as the residual anchor (as in NuResFormer), it faces conflicting objectives: its gate logits $G_1$ must perform effective, context-dependent selection for the first layer's own computation while also producing a reusable anchor signal $G_{\text{anc}}$ for all subsequent layers. The high first-layer gate activation suggests a resolution: the layer adopts a permissive gating policy to ensure the anchor gate logits retain broad, generally useful information, sacrificing some first-layer selectivity in the process.

ExoFormer exhibits an attenuated version of the same profile (Figure 8a); its first-layer activation is elevated compared to standalone gating but lower than NuResFormer's. As shown in Figure 8b, offloading the anchor role allows ExoFormer to partially restore the first layer's capacity for selective gating.

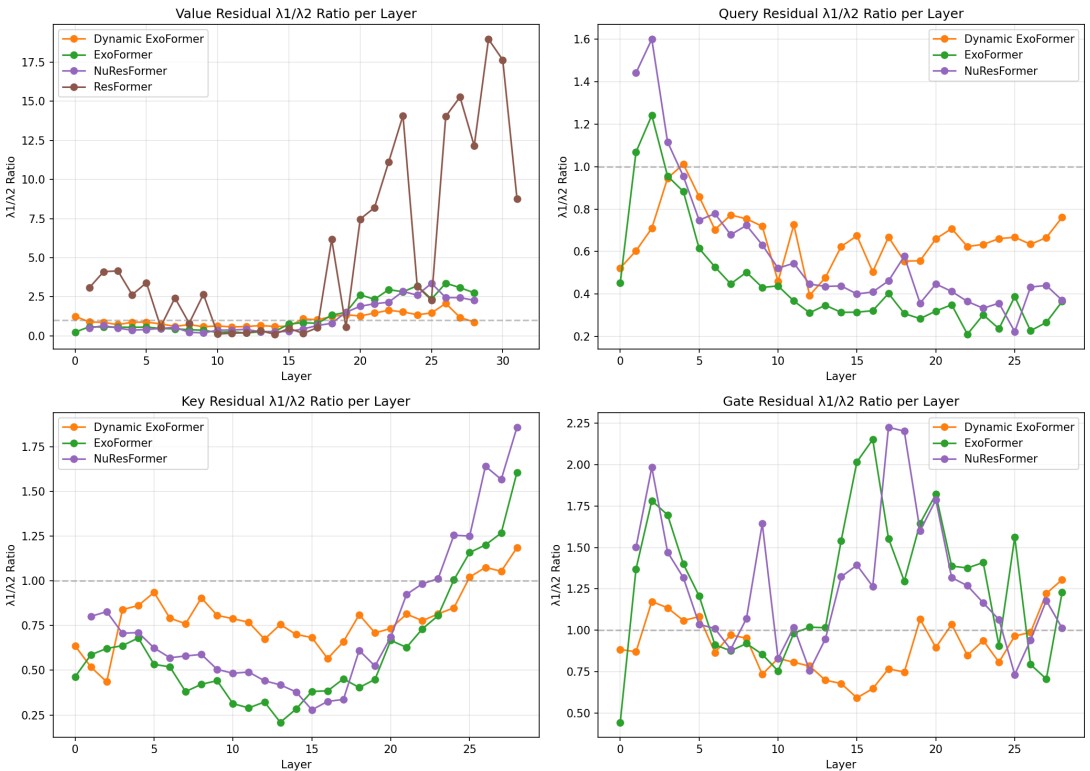

*Figure 7.* The ratio $\lambda_{n,1}/\lambda_{n,2}$ plotted for each component ($Q$, $K$, $V$, $G$) across layers for models using *elementwise* mixing for model variants $\sim 450M$. Values greater than 1 indicate stronger reliance on the anchor signal, while values less than 1 indicate preference for current-layer projections.

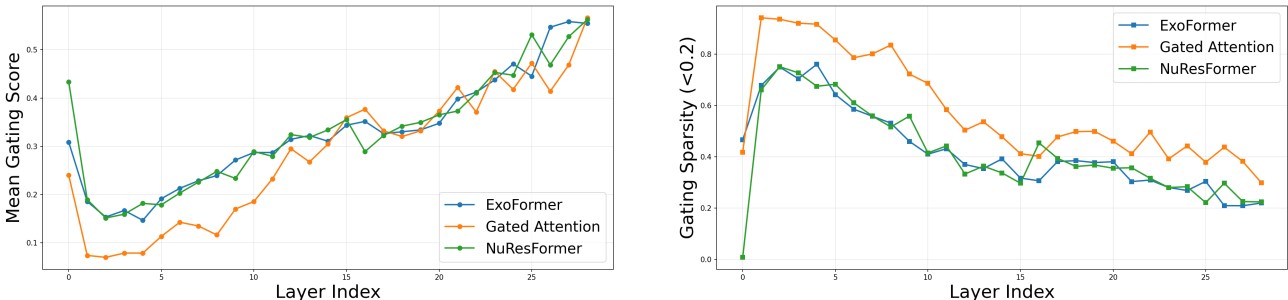

*(a)* Mean gating score ($\sigma(G)$) per layer. Higher values indicate less suppression of attention output.

*(b)* Gating sparsity per layer (% activations $< 0.2$). Higher sparsity indicates more selective, suppressive gating.

*Figure 8.* Analysis of gating behavior across model variants $\sim 450M$.

# C. Anchor Removal and Inverse Ablation Details

## C.1. Methodology

We perform two controlled inference-time ablations to isolate the causal role of the anchor under the Offloading Hypothesis.

**Anchor Removal ($\lambda_{n,1} = 0$).** For each layer $n$ and each component $S \in \{Q, K, V, G\}$, we zero out the anchor mixing coefficient while preserving the current-layer projection:

$$\widehat{S}_n = \lambda_{n,2}^S \odot S_n. \tag{13}$$

This tests whether sequential layers can maintain token identity without the anchor signal.

**Inverse Ablation ($\lambda_{n,2} = 0$).** Conversely, we zero out the current-layer contribution, forcing the model to rely exclusively on the normalized anchor:

$$\widehat{S}_n = \lambda_{n,1}^S \odot \mathrm{RMSNorm}(S_{\mathrm{anc}}). \tag{14}$$

This tests the fidelity of the anchor in isolation.

Both ablations are applied at inference time on the FineWeb-Edu validation set.

*Table 4.* Anchor ablation metrics for $\sim$450M models.

| MODEL | ABLATION | FINAL LAYER PCA | AVG. LAYER PCA |
|---|---|---|---|
| EXOFORMER | ANCHOR REMOVED ($\lambda_{n,1} = 0$) | 321 | 756 |
| NURESFORMER | ANCHOR REMOVED ($\lambda_{n,1} = 0$) | 181 | 685 |
| EXOFORMER | INVERSE ($\lambda_{n,2} = 0$) | 916 | 822 |
| NURESFORMER | INVERSE ($\lambda_{n,2} = 0$) | 577 | 611 |

As shown in Table 4, removing the anchor from ExoFormer reduces final-layer PCA core features to 321 dimensions (vs. $\sim$890 for the full model), while token-to-token similarity peaks at 93%. NuResFormer fares even worse, collapsing to 181 dimensions. This demonstrates that sequential layers in both architectures have offloaded identity preservation to the anchor, but NuResFormer's internal anchor is more brittle due to First-Layer Tension.

When forced to rely solely on the anchor, ExoFormer retains 822 PCA dimensions averaged across layers, compared to NuResFormer's 611. This gap confirms that the exogenous anchor preserves a higher-fidelity token identity signal than the internal anchor, which is compromised by its dual role.

In both Figure 9 and Figure 10, ExoFormer exhibits substantially higher robustness, with NuResFormer showing steeper collapse when the anchor is removed and lower feature retention when only the anchor is active.

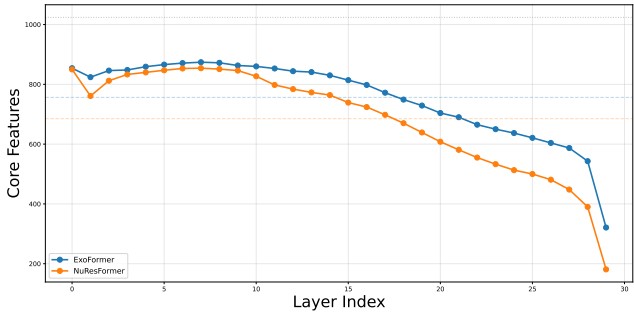

*Figure 9.* PCA core features (dimensions explaining 99% variance) across layers when the anchor is removed.

# D. Complexity Analysis

We present a complexity analysis of ExoFormer variants. Let $L$ be the number of Transformer layers and $d$ the model dimension ($d_{\mathrm{model}}$). We omit RMSNorm because it is negligible in terms of parameters and computation.

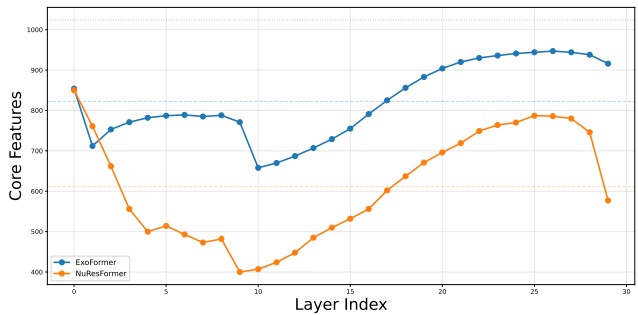

*Figure 10.* PCA core features (dimensions explaining 99% variance) across layers when only the anchor is used.

### D.1. Parameter Overhead

We analyze the parameter overhead of ExoFormer variants relative to a baseline Transformer with Gated Attention. Throughout, we ignore input and output embedding parameters, as they are shared across all models.

**Baseline Parameters:**   The baseline Transformer with Gated Attention has parameters per layer for the attention projections (queries, keys, values, gate logits and output) and the two-layer SwiGLU feed-forward network. The attention and output projections require $5d^2$ parameters (five $d \times d$ matrices), and the FFN requires $6d^2$ parameters (assuming expansion factor 4). Thus, ignoring the input and output embedding, the total parameters for $L$ layers are:

$$P_{\text{base}} = L \cdot \left(5d^2 + 6d^2\right) = 11Ld^2. \tag{15}$$

**Exogenous Anchor Parameters:**   ExoFormer introduces dedicated projection matrices for the exogenous anchor: four $d \times d$ matrices, one for each attention component (Q, K, V, G):

$$\Delta P_{\text{anchor}} = 4d^2. \tag{16}$$

**Static Mixing Parameters:**   For static mixing with elementwise granularity, each layer learns two mixing coefficients (one for anchor, one for current projection) for each of the four components. This amounts to $8d$ parameters per layer:

$$\Delta P_{\text{static}} = L \cdot 8d. \tag{17}$$

**Dynamic Mixing Parameters:**   The Dynamic Mixing (DM) module adds a small MLP per layer. With $d_{\text{DM}} = 16$ and $d_{\text{out}} = 8$, the dominant term is from the first weight matrix ($d \cdot d_{\text{DM}}$). The second weight matrix and biases (totaling 128 parameters) are negligible for typical $d$:

$$\Delta P_{\text{DM}} \approx L \cdot (d \cdot d_{\text{DM}}). \tag{18}$$

**Total Parameter Overhead:**   Static ExoFormer adds:

$$\Delta P_{\text{static ExoFormer}} = 4d^2 + 8Ld. \tag{19}$$

Dynamic ExoFormer additionally includes the DM module:

$$\Delta P_{\text{dynamic ExoFormer}} = 4d^2 + 8Ld + Ld \cdot d_{\text{DM}}. \tag{20}$$

**Parameter Ratio:**   The extra parameter ratio relative to baseline simplifies to:

$$R_P^{(\text{static})} = \frac{4d^2 + 8Ld}{11Ld^2} = \frac{4}{11L} + \frac{8}{11d}, \tag{21}$$

$$R_P^{(\text{dynamic})} = \frac{4d^2 + 8Ld + Ld \cdot d_{\text{DM}}}{11Ld^2} = \frac{4}{11L} + \frac{8 + d_{\text{DM}}}{11d}. \tag{22}$$

For values ($L = 32, d = 1024, d_{\text{DM}} = 16$), this corresponds to approximately 1.2% overhead for static ExoFormer and 1.3% for dynamic ExoFormer, a modest increase given the observed performance gains.

While these raw parameter counts suggest a negligible increase, they understate the effective capacity contribution of the anchor module: the exogenous parameters are structurally privileged because they are reused at every layer via the mixing framework. Consequently, ExoFormer trades a minimal increase in trainable parameters ($\sim$1.3%) for a significant gain in effective capacity by dedicating a module exclusively to identity preservation. This is more parameter-efficient than simply widening hidden dimensions, which distributes additional capacity indiscriminately across all layers.

### D.2. Computational Overhead (FLOPs)

We estimate the floating-point operations (FLOPs) per token during the forward pass, considering matrix-vector multiplications as the dominant factor ($2d^2$ FLOPs per $d \times d$ matrix). We include the cost of elementwise operations for mixing, although they are computationally smaller ($O(d)$) compared to projections ($O(d^2)$).

**Baseline FLOPs:**  The baseline usage corresponds to the parameters utilized at every layer. With $P_{\text{base}} = 11Ld^2$ (excluding embeddings), the computational cost per token is:

$$C_{\text{base}} \approx 2 \cdot P_{\text{base}} = 22Ld^2. \tag{23}$$

**Exogenous Anchor FLOPs:**  The exogenous anchor projections are computed only once per token using the input embeddings, regardless of the network depth. For the four projection matrices ($W_{\text{anc}}^Q, W_{\text{anc}}^K, W_{\text{anc}}^V, W_{\text{anc}}^G$), the cost is:

$$\Delta C_{\text{anchor}} \approx 2 \cdot (4d^2) = 8d^2. \tag{24}$$

Crucially, this cost is constant and does not scale with the number of layers $L$.

**Dynamic Mixing FLOPs:**  The Dynamic Mixing module operates at every layer. The computational cost includes the projection of the MLP ($d \to d_{\text{DM}}$) and the elementwise mixing operations. The mixing involves two multiplications (coefficients $\times$ anchor/current projections) and one addition per element for the four components (Q, K, V, G), totaling $12d$ FLOPs per layer:

$$\Delta C_{\text{DM}} \approx L \cdot (2 \cdot d \cdot d_{\text{DM}} + 12d). \tag{25}$$

**Total FLOPs Overhead:**  The total computational overhead ratio is:

$$R_{\text{FLOPs}} = \frac{\Delta C_{\text{anchor}} + \Delta C_{\text{DM}}}{C_{\text{base}}} = \frac{8d^2 + L(2dd_{\text{DM}} + 12d)}{22Ld^2} = \frac{8}{22L} + \frac{d_{\text{DM}}}{11d} + \frac{6}{11d}. \tag{26}$$

For values ($L = 32, d = 1024, d_{\text{DM}} = 16$), the overhead is dominated by the anchor projection ($\approx 1.1\%$), followed by the dynamic module projection ($\approx 0.14\%$), and the elementwise mixing operations ($\approx 0.05\%$), resulting in a total FLOPs increase of approximately 1.33%.

### D.3. Efficiency-Performance Trade-off

The FLOPs analysis presented above provides a lower-bound estimate of the computational overhead. In practice, the real-world slowdown may be higher (we observe an increase in latency per token of approximately 8–15% for our reference implementation prioritizing modularity and interpretability over speed, compared to the theoretical 1.3% FLOPs increase).

We emphasize that our primary contribution is a novel architecture that unifies attention projection mixing, not a production-optimized layer ready for deployment. The net performance benefit even in our setting, evidenced by accuracy gains of 1.5 points and data efficiency improvements of 1.5× against Gated Attention, remains positive even when accounting for the measured runtime overhead, suggesting that an optimized kernel implementation would close the gap with the theoretical 1.3% overhead.

# E. Alternative Normalization Placements

We begin from the E-ExoFormer (full residual norms) configuration as the baseline. All variants retain RMSNorm on the anchor sources before mixing (our default), but apply an *additional* normalization to the mixed tensors $\widehat{Q}_n, \widehat{K}_n, \widehat{V}_n, \widehat{G}_n$ under different schemes:

- **QKV-Norm.** We apply standard QKNorm-style RMS normalization to the mixed value tensor $\widehat{V}_n$ in addition to the existing QKNorm on $\widehat{Q}_n$ and $\widehat{K}_n$.

- **QKVG-Norm.** We apply RMSNorm to *all four* mixed tensors $(\widehat{Q}_n, \widehat{K}_n, \widehat{V}_n, \widehat{G}_n)$ after mixing but before attention computation and gating. To our knowledge, this specific four-way post-mixing normalization is unexplored in prior work.

Table 5 reports validation perplexity on FineWeb-Edu for the $\sim$450M parameter models after 10B tokens.

*Table 5.* Validation perplexity of E-ExoFormer under alternative normalization placements ($\sim$450M, 10B tokens). Lower is better.

| CONFIGURATION | VALIDATION PPL |
|---|---|
| E-EXOFORMER (BASELINE, PRE-INJECTION ONLY) | 14.13 |
| + QKV-NORM (POST-MIXING ON $\widehat{V}_n$) | 14.08 |
| + QKVG-NORM (POST-MIXING ON ALL FOUR) | 14.07 |

The gains from post-mixing normalization are marginal: QKV-Norm improves perplexity by only 0.05, and QKVG-Norm by 0.06, relative to the baseline. These small deltas suggest that normalizing the anchor sources *before* injection captures the vast majority of the stabilization benefit. The residual pathway already benefits from the distributional alignment provided by pre-injection RMSNorm (Section 4.2), and additional post-mixing normalization provides diminishing returns.

# F. Gradient Stability Analysis

Following the methodology of OLMo et al. (2025), we define the spike score as the percentage of gradient L2-norm values exceeding seven standard deviations from a rolling mean computed over the last 1,000 steps. A higher score indicates more severe gradient transients and less stable optimization.

*Table 6.* Gradient spike scores for 450M-parameter models. Lower is better.

| MODEL | SPIKE SCORE (%) |
|---|---|
| RESFORMER (VALUE RESIDUAL, UNNORMALIZED) | 0.0458 |
| NAÏVE COMBINATION (GATED + RESFORMER, UNNORMALIZED) | 0.0108 |
| GATED ATTENTION | 0.0085 |
| E-NURESFORMER (FULL RESIDUAL NORMS) | 0.0081 |
| E-EXOFORMER (FULL RESIDUAL NORMS) | **0.0054** |

ResFormer exhibits the highest spike score (0.0458), confirming that unnormalized value residuals introduce severe gradient instability. The naïve combination partially mitigates this (0.0108) but remains elevated relative to standalone Gated Attention, consistent with our hypothesis that $\sigma(G)$ is diverted from its intended filtering role to compensate for distributional mismatch. E-ExoFormer achieves the lowest score (0.0054), demonstrating that decoupling the anchor from the first layer eliminates the architectural tension as an independent source of gradient instability.

# G. Hyperparameters

*Table 7.* Training hyperparameters for models with different configurations for the main $\sim 450M$ models. Layer depth was reduced for gated variants to maintain comparable parameter counts. All models were trained on the FineWeb-Edu dataset.

| HYPERPARAMETER | NO GATE | WITH GATE | DECOUPLED (EXOFORMER) |
|---|---|---|---|
| PARAMETERS (M) | 454 | 453 | 457 |
| LAYERS | 32 | 29 | 29 |
| ATTENTION HEADS | | 16 | |
| HIDDEN DIMENSION | | 1024 | |
| FFN DIMENSION | | 4096 | |
| TIE WORD EMBEDDING | | FALSE | |
| VOCABULARY SIZE | | 57,601 | |
| ACTIVATION FUNCTION | | SWIGLU | |
| POSITION EMBEDDING | | ROPE ($\theta = 500{,}000$) | |
| SEQUENCE LENGTH | | 2048 | |
| BATCH SIZE (TOKENS) | | 262,144 | |
| TRAINING TOKENS | | 10B | |
| WARMUP STEPS | | 1000 | |
| WARMDOWN STEPS | | 7630 (20%) | |
| TOTAL STEPS | | 38,147 | |
| **OPTIMIZATION** | | | |
| OPTIMIZER | | MUON + ADAM | |
| MUON LEARNING RATE | | 0.01 | |
| ADAM LEARNING RATE | | 0.003 | |
| LEARNING RATE SCHEDULE | | LINEAR | |
| ADAM $\beta$ | | (0.9, 0.95) | |
| MUON MOMENTUM | | 0.95 | |
| GRADIENT CLIP | | 1.0 | |
| DROPOUT | | 0.0 | |
| CAUTIOUS WEIGHT DECAY | | TRUE | |
| MUON WEIGHT DECAY | | 0.1 | |
| ADAM WEIGHT DECAY | | 0.0 | |
| Z LOSS WEIGHT | | 1E-5 | |
| RMSNORM EPSILON | | 1E-6 | |
| QK NORMALIZATION | | TRUE | |

*Table 8.* Training hyperparameters for 1B parameter models. Most hyperparameters remain identical to those in Table 7 except for depth, width, and learning rates. Batch size was kept constant, requiring an increase in total steps to reach 20B training tokens.

| HYPERPARAMETER | GATED ATTENTION | DYNAMIC EXOFORMER |
|---|---|---|
| PARAMETERS (B) | 1.01 | 1.02 |
| LAYERS | 32 | |
| HIDDEN DIMENSION | 1536 | |
| FFN DIMENSION | 6144 | |
| **OPTIMIZATION** | | |
| MUON LEARNING RATE | 0.003 | |
| ADAM LEARNING RATE | 0.001 | |
| TRAINING TOKENS | 20B | |
| TOTAL STEPS | 76,293 | |

