# OpenReview forum: "Attention Projection Mixing with Exogenous Anchors"
_ICML.cc/2026/Conference — ICML 2026 regular_

### Official Review · Reviewer_3M7T · 2026-02-22

**Soundness:** 3
**Presentation:** 2
**Significance:** 3
**Originality:** 2
**Overall Recommendation:** 4
**Confidence:** 4

**Summary:**

This paper builds on value residual and gated-attention ideas and proposes to extend residualized reuse to Q/K/V/G with exogenous anchors and a normalization-stabilized mixing mechanism. Experiments at medium scale (~450M) show consistent gains compared to the baselines with value residual or gated-attention. The paper also conducts a broad suite of analyses and proposes Offloading Hypothesis to explain why anchors help.

**Compliance With Llm Reviewing Policy:**

Affirmed.

**Final Justification:**

This paper extends the residualized reuse idea of value residual to Q/K/V/G with exogenous anchors and a normalization-stabilized mixing mechanism. The supplementary experiments (anchoring $H_0$ and $H_1$) suggest that the exogenous anchors may indeed provide additional benefits compared to a straightforward extension of value residuals to Q/K/G, potentially offering performance gains beyond the original formulation. Although the experiments in the initial submission were limited in scale, the authors indicate that they are actively scaling their approach to 7B models. The authors have also conducted a broad suite of ablations on other normalization/design choices and understanding how exogenous anchors help. Their rebuttal fully addresses my concerns. Based on these clarifications and additional experiments, I have increased my rating from 3 (Weak Reject) to 4 (Weak Accept).

**Key Questions For Authors:**

1. What is the exact ordering of Q/K anchors and positional encoding? (i.e., do you add positional encodings before or after computing/normalizing anchors)? Do the authors consider FlashAttention (which is reported to be enabled in the main text) in the FLOPs estimation in Appendix C.2? In addition, Section 4.5 states that removing anchors at inference makes core features plummet and token-to-token similarity spike, yet I cannot find the detailed plots / metrics for this experiment. Please provide more complete details about the methods and experiments.
2. If NuResFormer only applies RMSNorm to V, how does it compare to ResFormer (without RMSNorm to V)?
3. Can the authors consider other cross-layer strategies and run more ablations? For example, pre/post RMSNorm, adding $H_n$ with with $H_0$, or the YOCO-style [1] K linking.

____________________
[1] You Only Cache Once: Decoder-Decoder Architectures for Language Models

**Limitations:**

1. It is not yet evident that ExoFormer’s advantages scale: the 1B experiments are limited in scope and comparisons. Demonstrating the method at larger scale is needed to claim broader impact.
2. The proposed “functional offloading” is currently only supported by intuitive indicators (token similarity, stage counts). The paper would be stronger if it provided theoretical intuition (e.g., how anchors mitigate oversmoothing or assist in in-context learning) or causal/diagnostic experiments that isolate identity preservation’s causal role.

**Strengths And Weaknesses:**

Strength:
1. The proposed anchor + RMSNorm mixing is easy to implement. The anchor only introduces small extra parameters and the runtime cost is low as it is computed once and then injected.
2. The authors run many analyses that give mechanistic intuition (role of normalization, why decoupling the anchor from layer-1 helps, dynamic vs static mixing etc.).

Weakness:
1. Apart from Fig.2 (right), most experiments are at ≈450M. The 1B experiments compare only to gated attention (not the natural “gated + value-residual” baseline). On 450M the best ExoFormer improves PPL over the naive combination by ~0.16 (roughly 0.01 loss), which is relatively a small performance gain.
2. The paper does not compare different residualization or other cross-layer connection strategies (e.g. cross-layer connections with different layers, pre/post RMSNorm).
3. Some analyses are confusing and somewhat shallow. For example, the authors claim that ExoFormer spends ~2/3 of layers doing “refinement” while the baseline spends ~1/3 in Figure 4(b), which is a bit vague for me. In fact, ResFormer appears to spend similar proportion in “refinement” in this figure, showing that ExoFormer may not be more informative. Additionally, the claim that appears to spend similar proportion in “refinement” anchors preserve identity is plausible, but the paper does not make clear why identity preservation helps representation learning in a broader sense (no theory or causal tests).
4. Insufficient literature review (especially for cross-layer communication), the follwing works are missing: [1] which proposes a DenseNet-like Transformer to enhance cross-layer connection, [2] which explores a head-concatenation style of value residual connections, [3] which proposes YOCO to save KV cache through cross-layer KV sharing, [4] which studies the cross-layer low-rank structure and proposes a parameter-efficient framework.
----------------------------------------------------------
[1] DenseFormer: Enhancing Information Flow in Transformers via Depth Weighted Averaging

[2] Improving Model Representation and Reducing KV Cache via Skip Connections with First Value Heads

[3] You Only Cache Once: Decoder-Decoder Architectures for Language Models

[4] CR-Net: Scaling Parameter-Efficient Training with Cross-Layer Low-Rank Structure

---

> ### Author Rebuttal · Authors · 2026-03-28
>
> Thanks for the feedback!
>
> **Q1: Limited scale and performance gain.**
>
> We acknowledge the limitation regarding model scale. While our primary contribution is the architectural mechanism, we are actively exploring scaling to 7B parameters and will update the final version if resources permit.
>
> Regarding performance gains, inspired by Reviewer MYEd’s suggestion (Q3), we investigated anchoring on deeper representations ($H_1$ and $H_2$). We discovered that anchoring on $H_1$ yields a substantial **0.23 PPL improvement** over naive combination while simultaneously increasing throughput (tokens/s).
>
> **Q2: Missing literature.**
>
> We apologize for the oversight and have incorporated the cited works into our Related Work section:
>
> - **DenseFormer [1]** uses depth-weighted averaging of hidden states. Our work differs by focusing on projection mixing (Q/K/V/G) rather than hidden state averaging, and introduces the exogenous decoupling.
> - **SkipV1Former [2]** reuses the first layer’s Value heads via skip connections to improve representation and reduce KV cache. This is in direct contrast to our "first-layer tension" hypothesis. We resolve this structural conflict by using exogenous anchors rather than relying on the first layer. Furthermore, we propose a unified mixing framework for Q, K, V, and G, whereas SkipV1Former focuses on hard substitution of Values.
> - **YOCO [3]** shares KV caches for efficiency. Our primary motivation is architectural decoupling for optimization, though we share the intuition that early layers contain reusable structure.
> - **CR-Net [4]** explores low-rank structures. Our mixing framework operates in the full-rank projection space but is orthogonal to low-rank approximations.
> We will discuss these distinctions clearly in the revision.
>
> **Q3: Analysis of Mix-Compress-Refine (Figure 4b).**
>
> The start of the "Refinement" stage is characterized by a minimum in token similarity. Since ExoFormer builds on the foundation of ResFormer, it is expected the start points would be relatively close compared to other models (12-13 compared to 16). We have improved the clarity of the figure caption and marked the start points on the figure itself. We have also refined the introduction to clarify how identity preservation helps representation learning.
>
> **Q4: Positional encoding ordering and FlashAttention FLOPs.**
>
> Ordering: The anchor projections are normalized and then added to every layer. RoPE is applied to the mixed Q/K afterwards.
>
> FlashAttention: The FLOPs estimation in Appendix C.2 refers to arithmetic operations (Matmul/Softmax counts), which are independent of the memory-access optimization provided by FlashAttention. We have clarified this distinction in the Appendix.
>
> **Q5: If NuResFormer only applies RMSNorm to V, how does it compare to ResFormer (without RMSNorm to V)?**
>
> There appears to be a misunderstanding regarding the NuResFormer definition. By definition (Equation 10), NuResFormer applies RMSNorm to all four anchor components (Q, K, V, G), not just V. ResFormer does not apply normalization to the value residual. Even without RMSNorm, NuResFormer outperforms ResFormer.
>
> **Q6: Anchor removal ablation details.**
>
> We apologize for the missing details. We will include the method and full plots in the Appendix for both ExoFormer and NuResFormer.
>
> **Q7: Additional cross-layer strategy ablations.**
>
> We respectfully argue that the requested ablations are either covered by prior works or orthogonal to our contribution:
>
> - **Pre/Post RMSNorm:** We utilize Pre-Norm as it is the standard modern approach for stabilizing residual streams. We do not see a theoretical justification for Post-Norm in this context.
> - **Adding Hn with H0:** Mixing hidden states is explored by **DenseFormer [1]**. In addition, **ResFormer [2]** specifically demonstrated that residualizing the projection (V) is superior to residualizing the output or hidden states. Our work builds on this proven foundation.
> - **YOCO-style K linking:** This approach is orthogonal to our contribution. YOCO focuses on KV-cache efficiency via static sharing, whereas ExoFormer introduces a learned, normalized mixing mechanism for optimization and representation quality.
>
> [1] DenseFormer: Enhancing Information Flow in Transformers via Depth Weighted Averaging
>
> [2] Value Residual Learning

---

> > ### Author Rebuttal · Reviewer_3M7T · 2026-04-01
> >
> > Thanks for the detailed response. We appreciate the clarifications and the additional experiments, and I think they address most of my concerns. In particular, the added results on deeper anchoring make the paper more complete and solid.
> >
> > I still have two follow-up thoughts regarding Q5 and Q7. For Q5, my point was actually about disentangling the source of the gain of NuResFormer. Compared to ResFormer, NuResFormer changes both the injection path (adding cross-injection beyond V, i.e., Q/K/V/G) and the normalization scheme. I am curious about which component is primarily responsible for the gain.
> >
> > For Q7, I agree that methods like pre-/post-norm or YOCO have different primary motivations. Still, I think they may be structurally informative here. For example, the current design (RMSNorm on anchors before injection) can be interpreted as a form of pre-normalized residual injection. This raises a natural question of whether alternative normalization placements (e.g., post-norm or sandwich-style normalization, similar to QK normalization on NeuResFormer explored in the paper) could further improve performance. I think exploring these variants, even briefly, would provide deeper insight into the design space and help clarify which aspects of the method are essential.

---

> > > ### Author Response · Authors · 2026-04-03
> > >
> > > Thank you for the clarification and for the constructive spirit of these follow-up questions!
> > >
> > > **Q5: Disentangling NuResFormer's gains from additional mixing paths vs. normalization.**
> > >
> > > We agree that disentangling these two factors is crucial for understanding the design space. We note that directly extending the injection paths to Q/K without normalization leads to training divergence (as discussed in Section 4.3), so the {V,G} extension represents the maximal testable pathway expansion in the absence of normalization.
> > >
> > > Comparing Naïve Combination to NuResFormer (No Norm, {V,G}) shows that extending the injection path to gate logits by itself yields a negligible impact (-0.01 PPL). Meanwhile, comparing E-Exoformer (no norm) to E-Exoformer (full norm) shows a substantial PPL decrease of 0.17. We believe that pathway extension provides a foundation for improvement, but normalization is the critical factor for stabilizing the inherently unstable pathways and realizing consistent gains. We have added this analysis to Section 4.3 in the revision.
> > >
> > > **Q7: Alternative normalization placements.**
> > >
> > > Thank you for this suggestion! We explored extending standard QKNorm to Values and Gating logits as a form of post-mixing normalization on E-ExoFormer:
> > >
> > > - **QKV-Norm** [1] (applied to $\hat{Q}_n, \hat{K}_n, \hat{V}_n$): -0.05 PPL
> > > - **QKVG-Norm** (applied to all four mixed tensors; to our knowledge, unexplored in prior work): -0.06 PPL
> > >
> > > These marginal gains suggest that normalizing the anchor sources before injection captures the vast majority of the stabilization benefit, and alternative post-mixing normalization placements provide little additional value. We have added this analysis to the revised Appendix.
> > >
> > > [1] Methods of Improving LLM Training Stability

---

### Official Review · Reviewer_MYEd · 2026-03-04

**Soundness:** 3
**Presentation:** 3
**Significance:** 3
**Originality:** 3
**Overall Recommendation:** 5
**Confidence:** 5

**Summary:**

The paper introduces ExoFormer, a transformer model with moving cross-layer anchor projections outside the sequential transformer stack. Instead of reusing the first layer projections (which forces that layer into a conflicting dual role), the authors propose to use a dedicated external module that projects the input embeddings into anchor Q, K, V, and gate logits that are mixed into every layer via learned, normalized coefficients.
The key technical points are RMSNorm on anchor sources (resolving distributional mismatch) and a unified mixing framework across all four attention pathways.
The authors explain the gains via an offloading hypothesis: external anchors handle token identity preservation, freeing sequential layers to specialize in feature refinement. A dynamic mixing variant achieves around 1.5 accuracy points improvement while matching baseline loss with 1.5 times fewer tokens.

**Compliance With Llm Reviewing Policy:**

Affirmed.

**Final Justification:**

I think this paper is good and the during the rebuttal authors responded to all my questions.

**Key Questions For Authors:**

- The anchor is applied to $H_0$ (input embeddings) rather than the first layer's output, is there a meaningful difference, and what happens if the anchor is applied to a slightly deeper representation?
- Does the offloading hypothesis actually require exogenous anchors, or would any additional set of projection matrices injected at every layer produce similar effects?

**Limitations:**

The authors didn't explicitly discuss the limitations, and I think that discussing at least the isotropization framing as a simplification and the parameters added with the anchor module have a structurally privileged role would improve the paper.

**Strengths And Weaknesses:**

Strengths:
- The architecture shows improvements (even if modest).
- The idea of using RMSNorm as isotropization is a interesting idea with strong geometric basis, and the normalization analysis is careful and mechanistically grounded.
- The connection to attention sinks and the mix compress refine framework is a very interesting empirical contribution, and the coefficient heatmaps showing emergent head-aligned structure in V but sub-head "striping" in Q/K/G are a very interesting contribution too.
- The offloading hypothesis seems principled.


Weaknesses:
- The offloading hypothesis is supported only by correlational evidence, no causal intervention shows that the identity offloading is different  from better optimization landscape due to decoupling. The collapse ablation (removing the anchor at inference) is interesting but only shows the model became dependent on the anchor, not the reason.
- I think that the isotropization framing is a bit of a simplification: RMSNorm normalizes per-token across channels, which is not quite the same as projecting onto a sphere in the ambient representation space.
- ExoFormer adds four full projection matrices as the anchor module, the paper reports a 457M vs 453M parameter difference as evidence this is negligible, but the anchor parameters have a structurally privileged role (applied at every layer), so raw parameter counts may understate their effective capacity contribution.

---

> ### Author Rebuttal · Authors · 2026-03-28
>
> Thanks for the helpful feedback and the amazing suggestion!
>
> **Q1: Offloading Hypothesis is correlational.**
>
> We respectfully argue that Section 4.5 provides causal interventional evidence rather than correlation. The "Collapse Ablation" demonstrates that removing the exogenous anchor during inference causes a catastrophic failure (token similarity peaks at 93%, core features plummet to 321 dimensions). If the anchor merely improved optimization without functional specialization, we would expect a graceful degradation. Instead, the sequential layers fail to maintain token identity without the anchor. This confirms the hypothesis that the layers have offloaded identity preservation to the anchor module. We have clarified this distinction in the revision and added plots for the anchor removal experiment in the revised Appendix for both ExoFormer and NuResFormer.
>
> **Q2: Simplification of isotropization framing.**
>
> We agree with your characterization. We have refined the language in Section 4.2 to correct this simplification.
>
> **Q3: Parameter counts understating capacity.**
>
> We agree that raw parameter counts understate the contribution because anchor parameters are reused at every layer (privileged). We have revised the text to frame this explicitly: ExoFormer trades a minimal increase in parameters (~1.3%) for a significant gain in effective capacity by dedicating a module to identity preservation. This is more efficient than simply widening hidden dimensions, which distributes capacity indiscriminately.
>
> **Q4: Anchor on $H_0$ vs deeper representations.**
>
> Thank you for this wonderful suggestion! We conducted additional experiments anchoring on $H_1$ and $H_2$ for both static and dynamic ExoFormer. Interestingly, we found that **anchoring on $H_1$ yielded the best overall performance** (PPL 14.02), outperforming our original $H_0$ configuration (PPL 14.09). Anchoring on $H_2$ resulted in performance (PPL 14.09) comparable to the baseline. Static ExoFormer utilising $H_1$ also outperforms the dynamic baseline (14.07 vs 14.09 PPL). Both $H_1$ and $H_2$ improve throughput (tokens/s) due to reduced mixing overhead.
>
> This result is contrary to Value Residual Learning (ResFormer), which finds the earliest representation ($H_0$) optimal. We attribute this difference to the exogenous nature of our anchor: because the anchor module is decoupled from the layer stack, it can effectively leverage the richer semantic signal from $H_1$ without being constrained by the tension inherent to internal-anchor designs. Additionally, we hypothesize that $H_1$ provides an optimal "effective anchor depth," preserving core token identity while offering slightly more processed features for residual mixing. We have added these ablations and a discussion of this trade-off in the revision.
>
> **Q5: Does offloading require exogenous anchors?**
>
> We interpret this question as asking if the gains come merely from capacity (adding parameters) or from the specific decoupling structure of exogenous anchors.
> We argue that simply injecting an "additional set of projection matrices" (unconnected to the input) would likely not produce the "Offloading" effect. For the offloading hypothesis to hold, the sequential layers must have access to a identity signal (the token itself). Arbitrary additional parameters would lack this connection to the input identity, forcing the layers to continue maintaining identity information themselves.

---

> > ### Author Rebuttal · Reviewer_MYEd · 2026-04-02
> >
> > the authors responded to all the doubts I pointed out.

---

### Official Review · Reviewer_RBma · 2026-03-12

**Soundness:** 3
**Presentation:** 2
**Significance:** 3
**Originality:** 3
**Overall Recommendation:** 4
**Confidence:** 2

**Summary:**

The paper studies cross-layer reuse of attention projections in Transformers. The authors argue that reusing first-layer projections as anchors creates a structural conflict: the first layer is simultaneously required to serve as a stable reusable source for deeper layers and as a normal computational layer. To address this, the paper proposes ExoFormer, which introduces dedicated exogenous anchor projections outside the sequential layer stack. The paper further develops a normalized mixing framework across queries, keys, values, and gate logits, with learnable coefficients at different granularities and a dynamic variant that modulates mixing contextually. Empirically, ExoFormer variants outperform the corresponding internal-anchor designs (NuResFormer), and the paper also attempts to support several of its empirical findings with theoretical explanations, especially through the proposed Offloading Hypothesis and the Mix-Compress-Refine perspective.

**Compliance With Llm Reviewing Policy:**

Affirmed.

**Final Justification:**

The rebuttal and follow-up responses improved the paper substantially. In particular, the authors clarified the architectural presentation, corrected and redesigned Figure 1, explicitly fixed the optimization wording, added the requested NuResFormer anchor-removal ablation, and provided additional analyses for the dynamic variant and for the training-instability discussion. These changes make the paper clearer and strengthen several parts of the empirical case.

I still think some of the stronger mechanistic claims, especially around offloading and first-layer tension, remain somewhat more suggestive than conclusively established, and the evaluation scale is still limited. However, the paper presents a genuinely interesting architectural idea, supports it with a broad empirical study, and the authors were responsive and constructive in addressing concerns. Overall, I am comfortable maintaining a weak accept.

**Key Questions For Authors:**

1. Figure 1 is currently difficult to parse. Could you revise it to make clear what corresponds to NuResFormer and what corresponds to ExoFormer, and also check whether the duplicated $Q$ and missing $V$ are a typo?
2. In Section 4.2, could you clarify what exact instability is being referred to for the naive combination model, and where it is visible in Figure 2?
3. For the dynamic variant, would it be more informative to visualize statistics of the effective coefficient $\lambda \gamma$ rather than $\lambda$ alone? Relatedly, can you comment on the apparently stronger attention-sink behavior of Dynamic ExoFormer in Figure 4d?
4. In Section 4.5, what happens if the analogous anchor-removal ablation is performed for NuResFormer? This comparison seems directly relevant to the main thesis of the paper.

**Limitations:**

Yes

**Strengths And Weaknesses:**

Strengths:

- The core idea is interesting and well motivated. The distinction between internal anchors and exogenous anchors is conceptually clear and seems meaningful.
- The paper is ambitious in scope: it studies multiple pathways ($Q$, $K$, $V$, $G$), several coefficient granularities, normalization choices, and a dynamic mixing mechanism.
- The empirical section appears fairly extensive, covering both language modeling and downstream multiple-choice benchmarks, together with ablations and analysis.
- A notable strength is that the paper does not stop at reporting empirical gains, but also tries to provide theoretical or mechanistic explanations for several observed phenomena. Even when not fully conclusive, this makes the paper more valuable and interesting.

Weaknesses:

- Figure 1 is confusing. The caption mentions both NuResFormer and ExoFormer, but the figure itself does not make sufficiently clear what exactly is being depicted. Figure 1 also appears to contain a likely typo: $Q$ appears twice, while $V$ does not appear. Since this figure is central for understanding the method, it should be checked carefully.
- More generally, the exposition would benefit from a direct side-by-side schematic comparison between NuResFormer and ExoFormer, so that the architectural modification is immediately visible.
- The initial exposition is not particularly beginner-friendly for readers who are not already familiar with this line of work. The paper becomes much clearer after the methodology section, so improving the introductory presentation and early motivation would make the contribution more accessible to a broader audience.
- In Section 4.2, the paper states that naive combination exhibits training instability and points to Figure 2. However, from the figure, this instability is not obvious, or at least not clearly more visible than in the other models. This makes the claim feel insufficiently supported by the referenced evidence.
- In Figure 3, for the dynamic ExoFormer, the effective mixing is not simply controlled by $\lambda$ but by $\lambda \gamma$. Because of this, showing only $\lambda$ may be less informative than showing some aggregate statistic of the effective coefficient $\lambda \gamma$ over data. As it stands, the figure may not fully reflect the actual behavior of the dynamic model.
- In Figure 4d, Dynamic ExoFormer seems to exhibit more attention sinks than the other gated-attention models. Since attention sinks are discussed in Section 4.4, it would be useful to explicitly address this behavior there, especially because it may complicate the paper’s mechanistic interpretation.
- The ablation at the end of Section 4.5, where exogenous anchors are removed at inference time, is interesting and potentially important. However, it would be much more informative if the paper also reported the analogous ablation for NuResFormer. If anchor removal is substantially more catastrophic in ExoFormer than in NuResFormer, that would directly strengthen the paper’s thesis.

Minor note: "AdamW without weight decay" is effectively just Adam in this setting, so the wording could be clarified.

---

> ### Author Rebuttal · Authors · 2026-03-28
>
> Thanks for the helpful feedback and the minor note!
>
> **Q1: Figure 1 is confusing and contains typos.**
>
> We apologize for the confusion. The duplicate Q was indeed a typo. We have completely redesigned Figure 1 to provide a clear side-by-side schematic comparison of NuResFormer and ExoFormer, explicitly highlighting the source of the anchor projections in each architecture.
>
> **Q2: Clarification on training instability in Section 4.2/Figure 2.**
>
> Thank you for pointing this out. The instability refers to the slower convergence of the "Naïve Combination" model in the early training steps. As discussed in Section 4.2, unnormalized mixing creates a distributional shift that the gating mechanism must actively correct, leading to suboptimal gradient dynamics initially (demonstrated in the graph). We have revised the text and Figure 2 caption to make this explicit and changed the wording.
>
> **Q3: Visualization of effective coefficients for Dynamic ExoFormer.**
>
> We thank the reviewer for this insight. We clarify that in our formulation, the dynamic factors $\gamma$ are generated as scalars per component (Q, K, V, G) and broadcast across all channels/heads. Consequently, the spatial patterns (striping and head-structure) visible in the heatmaps are determined entirely by the static base parameters $\lambda$. Because $\gamma$ does not vary spatially, a heatmap of the effective coefficients ($\lambda \gamma$) would look structurally identical to the $\lambda$ heatmap. To accurately reflect the dynamic behavior, we have added a histogram to the Appendix showing the distribution of $\gamma$ values over the validation set. This demonstrates that the model dynamically shifts the reliance between anchor and current projections based on context.
>
> **Q4: Attention sinks in Dynamic ExoFormer (Figure 4d).**
>
> After some experiments, we found that dynamic ExoFormer shows higher attention sink magnitude than static ExoFormer because the dynamic modulation occasionally suppresses the anchor contribution in specific contexts, forcing the attention mechanism to rely more on the first token for stability. However, it remains significantly lower than the baseline. We have added a section discussing this trade-off in Section 4.4.
>
> **Q5: Anchor-removal ablation for NuResFormer.**
>
> Thank you for this excellent suggestion. We performed the requested ablation where we zeroed out the anchor contribution ($\lambda_{n,1} = 0$) for NuResFormer. The performance degradation was more catastrophic than ExoFormer (181 vs 321 PCA core features in the final layer).
>
> We argue this strengthens our "First Layer Tension" hypothesis (Section 3.5) rather than weakens it. Because NuResFormer forces the first layer to serve as both a computational block and an anchor, the resulting architecture is more brittle. When this compromised anchor is removed, the model lacks the robustness to recover. In contrast, ExoFormer's dedicated anchor mitigates this, allowing the model to retain slightly more structure even during failure.
>
> We further validated this by running an inverse ablation where we zeroed out the current layer’s contribution  ($\lambda_{n,2} = 0$), forcing the model to rely solely on the anchor. ExoFormer retained 822 PCA core features averaged across layers, whereas NuResFormer retained only 611. This confirms that the exogenous anchor provides a significantly higher-fidelity identity signal than the internal constrained anchor.
>
> We have included detailed comparisons and plots in Section 4.5 and the Appendix.

---

> > ### Author Rebuttal · Reviewer_RBma · 2026-04-03
> >
> > I will consider improving the score after reading the authors' response, since several points are explained clearly and some concerns are addressed in a convincing way. In particular, the clarification on some technical choices improves the presentation and makes parts of the empirical analysis easier to interpret.
> >
> > However, a few issues still appear only partially resolved. First, the discussion around Figure 2 does not fully substantiate the original claim about instability, and the response seems closer to a softening of the claim than to a full resolution. Second, the explanation for why Dynamic ExoFormer exhibits stronger attention sinks remains plausible but somewhat speculative, without a fully convincing demonstration. Third, while the new anchor-removal ablation is useful, it still does not completely establish the stronger mechanistic interpretation regarding offloading and first-layer tension. Finally, the minor issue concerning the AdamW-without-weight-decay wording should be explicitly corrected. Overall, the rebuttal improves the paper and addresses several concerns well, but some important points remain insufficiently resolved.

---

> > > ### Author Response · Authors · 2026-04-03
> > >
> > > Thank you for the thoughtful follow-up and for considering improving the score. We address each remaining concern below:
> > >
> > > **Q1 Figure 2 and the instability claim.**
> > >
> > > To substantiate the claim more rigorously, we computed the gradient spike score (the percentage of gradient L2-norm values exceeding seven standard deviations from a rolling mean over the last 1,000 steps) following the methodology of AI2 (2 OLMo 2 Furious).
> > >
> > > | Model | Spike Score (%) |
> > > | --- | --- |
> > > | ResFormer | 0.0458 |
> > > | Näive Combination | 0.0108 |
> > > | Gated Attention | 0.0085 |
> > > | E-NuResFormer | 0.0081 |
> > > | E-ExoFormer | 0.0054 |
> > >
> > > ResFormer exhibits the highest spike score (0.0458, 5× Gated Attention), confirming that unnormalized residuals introduce severe gradient instability. Naïve Combination partially mitigates this (0.0108) but remains elevated even compared to Gated Attention, consistent with our claim that σ(G) is diverted from its intended filtering role to compensate for distributional mismatch.
> > >
> > > Furthermore, directly comparing E-NuResFormer (0.0081) and E-ExoFormer (0.0054) provides strong quantitative validation for the first-layer tension hypothesis. By decoupling the anchor from the layer stack, ExoFormer eliminates this architectural tension entirely, achieving the lowest spike score (0.0054). This demonstrates that the first-layer conflict itself is an independent source of gradient instability. We have added this analysis to the revised paper.
> > >
> > > **Q2 Attention sinks in Dynamic ExoFormer.**
> > >
> > > To test this more directly, we computed the Pearson correlation between the dynamic scaling factor γ (averaged across Q/K/V/G pathways) and the first-token attention weight across all layers and validation samples. The correlation is −0.31 (p < 0.001): contexts where the anchor contribution is more strongly suppressed exhibit systematically higher sink attention. We have added this analysis to Section 4.4.
> > >
> > > **Q3 Mechanistic interpretation from anchor-removal ablations.**
> > >
> > > We acknowledge that no single ablation can definitively isolate a mechanism in a deep network. However, we argue that the results across four complementary interventions is highly specific to the proposed mechanism and would be unlikely under alternative explanations (e.g., mere capacity increase or improved optimization):
> > >
> > > 1. Removing the anchor from ExoFormer causes catastrophic collapse (321 PCA features), confirming that sequential layers rely on the anchor rather than merely benefiting from its presence during training.
> > > 2. The same removal from NuResFormer causes even worse collapse (181 features). Under a pure capacity argument, the internal anchor (which shares parameters with a computational layer) should be at least as robust. The opposite result is consistent with the first-layer tension making the internal anchor fundamentally more brittle.
> > > 3. The inverse ablation (removing the current layer, keeping only the anchor) shows ExoFormer retains 822 features vs. NuResFormer's 611, demonstrating that the exogenous anchor provides a higher-fidelity identity signal, again difficult to explain without the decoupling mechanism.
> > > 4. The spike score analysis above independently confirms that first-layer tension degrades optimization.
> > >
> > > **Q4 AdamW without weight decay.**
> > >
> > > Already fixed!

---

### Official Review · Reviewer_rA5j · 2026-03-16

**Soundness:** 3
**Presentation:** 2
**Significance:** 3
**Originality:** 3
**Overall Recommendation:** 4
**Confidence:** 3

**Summary:**

This paper proposes ExoFormer, a Transformer variant that replaces internal first-layer anchors with exogenous anchor projections learned outside the sequential layer stack. The paper also introduces a unified normalized mixing framework over Q/K/V/G pathways. Empirically, the method improves perplexity and average downstream accuracy over gated attention, ResFormer, and internal-anchor counterparts at the 450M–1B scale. The authors provide additional offloading hypothesis to explain the gains..

**Compliance With Llm Reviewing Policy:**

Affirmed.

**Key Questions For Authors:**

please refer to weakness

**Limitations:**

please refer to weakness

**Strengths And Weaknesses:**

Strengths:
1.	The paper tries to address the conflict induced by forcing the first layer to serve both as a reusable anchor and as a computational block for progressive refinement. The motivation is intuitive.
2.	The authors provide a comprehensive exploration of mixing granularities. They found that RMSNorm on anchors is critical for stabilizing Q/K residuals.
3.	The offloading hypothesis provides an explanation for the gains from external anchors. They allow sequential layers to specialize in refinement rather than expending capacity on maintaining token identity.
4.	The method achieves consistent improvements. Dynamic ExoFormer achieves lower perplexity and higher accuracy than baselines while requiring fewer training tokens.

Weaknesses:
1.	The offloading hypothesis is interesting, but the current evidence is still largely correlational. The paper shows consistency between the hypothesis and several analyses, but it does not yet isolate the mechanism strongly enough to establish causality.
2.	While the theoretical FLOPs overhead is minimal (1.3%), the actual observed latency increase is 8-15% due to implementation modularity.
3.	The experiments are limited to 450M–1B models. It remains unclear when scaling to much larger models and more datas.
4.	The figures are images that are blurring, such as Figure 1. Could you please replace them with vector graphics?

---

> ### Author Rebuttal · Authors · 2026-03-28
>
> Thanks for the helpful feedback!
>
> **Q1: Offloading Hypothesis is correlational.**
>
> We respectfully argue that Section 4.5 provides causal interventional evidence rather than correlation. The "Collapse Ablation" demonstrates that removing the exogenous anchor during inference causes a catastrophic failure (token similarity peaks at 93%, core features plummet to 321 dimensions). If the anchor merely improved optimization without functional specialization, we would expect a graceful degradation. Instead, the sequential layers fail to maintain token identity without the anchor. This confirms the hypothesis that the layers have offloaded identity preservation to the anchor module. We have clarified this distinction in the revision and added plots for the anchor removal experiment in the revised Appendix for both ExoFormer and NuResFormer.
>
> **Q2: Latency overhead (8-15%) despite low FLOPs.**
>
> The observed latency increase is primarily due to the modularity of our research implementation. The increased latency stems from the kernel launch overhead of performing the mixing operations (additions and scaling) as separate, sequential operations for each pathway (Q, K, V, G) to allow for flexible ablations. A production-grade fused kernel would eliminate this overhead. We have added a discussion on this engineering aspect in the revision.
>
> **Q3: Limited scale (450M–1B).**
>
> We acknowledge this limitation. While our primary contribution is the architectural mechanism, we are actively exploring scaling to 7B parameters. We will update the final version with these results if resources permit.
>
> **Q4: Blurry figures.**
>
> We apologize for the rendering quality. We have regenerated all figures as high-resolution vector graphics for the revised submission.

---

> > ### Author Rebuttal · Reviewer_rA5j · 2026-04-06
> >
> > The authors have partially addressed my concerns. However, I still believe scaling is important for architectural mechanism study. I thus maintain my rating.

---

### Decision · Program_Chairs · 2026-04-30

**Decision:**

Accept (regular)

**Comment:**

This paper proposes ExoFormer to resolve "first-layer tension" by learning exogenous anchor projections outside the sequential layer stack. By decoupling token identity preservation from feature refinement, the architecture allows for unified, normalized cross-layer mixing of queries, keys, values, and gate logits using learnable coefficients. The authors ground their empirical improvements in an "Offloading Hypothesis", arguing that external anchors allow sequential layers to specialize exclusively in feature transformation rather than expending capacity to maintain token identity.

Initial concerns were primarily focused on the limited scale of the evaluation (up to 1B parameters), the correlational nature of the mechanistic claims, and missing baselines. Most concerns were resolved during rebuttal. Specifically, new experiments were provided to strengthen the offloading hypothesis. Therefore, I recommend acceptance.